TOPICAL REVIEW

# Motor potentials evoked by transcranial magnetic stimulation: interpreting a simple measure of a complex system

Danny Adrian Spampinato[1,2,3] (ID), Jaime Ibanez[1,4,5] (ID), Lorenzo Rocchi[1,6] (ID) and John Rothwell[1]

[1]Department of Clinical and Movement Neurosciences, University College London, London, UK
[2]Department of Human Neurosciences, Sapienza University of Rome, Rome, Italy
[3]Department of Clinical and Behavioral Neurology, IRCCS Santa Lucia Foundation, Rome, Italy
[4]BSICoS group, I3A Institute and IIS Aragón, University of Zaragoza, Zaragoza, Spain
[5]Department of Bioengineering, Centre for Neurotechnologies, Imperial College London, London, UK
[6]Department of Medical Sciences and Public Health, University of Cagliari, Cagliari, Italy

Handling Editors: Laura Bennet & Richard Carson

The peer review history is available in the Supporting Information section of this article (https://doi.org/10.1113/JP281885#support-information-section).

**Abstract** Transcranial magnetic stimulation (TMS) is a non-invasive technique that is increasingly used to study the human brain. One of the principal outcome measures is the motor-evoked potential (MEP) elicited in a muscle following TMS over the primary motor

**Danny Adrian Spampinato** received his PhD in Biomedical Engineering from Johns Hopkins University (Baltimore, USA) in 2017. He has held a research associate position at the Department of Clinical and Movement Neurosciences at the University College London (London, UK). Currently, he is a senior research fellow at the Non-Invasive Brain Stimulation Unit/Department of Behavioural and Clinical Neurology at Fondazione Santa Lucia Foundation IRCCS (Rome, Italy) and at 'Sapienza' University of Rome under the Department of Human Neuroscience. His research concentrates on understanding the physiology and pathophysiology of human movement control in health and neurological disease and he is interested in the development of non-invasive brain stimulation methods.

cortex (M1), where it is used to estimate changes in corticospinal excitability. However, multiple elements play a role in MEP generation, so even apparently simple measures such as peak-to-peak amplitude have a complex interpretation. Here, we summarize what is currently known regarding the neural pathways and circuits that contribute to the MEP and discuss the factors that should be considered when interpreting MEP amplitude measured at rest in the context of motor processing and patients with neurological conditions. In the last part of this work, we also discuss how emerging technological approaches can be combined with TMS to improve our understanding of neural substrates that can influence MEPs. Overall, this review aims to highlight the capabilities and limitations of TMS that are important to recognize when attempting to disentangle sources that contribute to the physiological state-related changes in corticomotor excitability.

(Received 31 August 2022; accepted after revision 18 May 2023; first published online 30 March 2023)

**Corresponding author** D. Spampinato: Department of Clinical and Movement Neurosciences, University College London, London, UK. Email: d.spampinato@ucl.ac.uk

**Abstract figure legend** Transcranial magnetic stimulation (TMS) is a fundamental tool to non-invasively study the human brain. The motor-evoked potential (MEP) elicited in a muscle following TMS over the primary motor cortex (M1) is one of the most used measures for non-invasive quantification of cortical and spinal excitability in humans. However, given the multiple elements playing a role in MEP generation and modulation, its interpretation can be highly complex. This review summarizes what is currently known regarding the main cortical neurons, connections and circuits contributing to modulations in MEP amplitudes, and we also discuss relevant concepts to be considered when interpreting MEPs measured across different brain states and patient populations.

## Introduction

Transcranial magnetic stimulation (TMS) is a common non-invasive technique used to study the physiology of the human brain, particularly in the motor system where its ability to evoke an immediate and measurable neuronal response makes it easy to integrate into both basic and clinical research settings. However, ease of use conceals the complexities involved when TMS interacts with cortical neurons (Rothwell et al., 1991; Ziemann et al., 1996; Di Lazzaro et al., 1998).

The commonest outcome measure in such studies is the motor-evoked potential (MEP), the electromyographic (EMG) correlate of the muscle twitch evoked by delivering TMS over the contralateral primary motor cortex (M1) (Barker et al., 1985; Rothwell et al., 1987). Yet although the MEP is evoked from M1, its amplitude, threshold and latency depend on activity in circuits of both cortex and spinal cord (Di Lazzaro & Rothwell, 2014). Indeed, even the recruitment order and synchronicity of motor units in muscles play an important role.

Here, we begin with a brief overview of the neural circuits involved in the generation of MEPs and introduce how adjusting stimulation parameters can extract information regarding the activity of specific circuits contributing to the MEP. We then provide some examples of using MEPs in the context of motor preparation and in different pathological conditions, with the aim to provide some perspectives for interpreting corticospinal excitability measurements based on TMS.

Finally, we discuss how the combination of TMS with electroencephalography (EEG) may represent a possible strategy to probe cortical motor excitability without the confounding factor represented by the activation of spinal motor circuitry.

## Basic principles of MEPs elicited by M1 TMS

**Magneto-electrical stimulation.** The magnetic field produced by a TMS pulse ($\sim$100 $\mu$s duration) penetrates the scalp and skull. Because it is a time-varying field, it induces an electric current (100–200 $\mu$s duration) in the brain with an intensity proportional to the rate of change of the magnetic field (Barker, 1991). Magnetic field strength falls off rapidly with distance from the TMS coil surface so that a typical figure-of-eight coil used in human studies can activate neurons up to $\sim$2–3 cm from the cortical surface (Deng et al., 2013). Sufficiently strong currents are thought to depolarize neurons preferentially at presynaptic boutons (and at axonal bends or at points where the external resistance changes such as the grey–white matter interface) where they generate an action potential that can spread antidromically back towards the cell body and orthodromically to activate the synapse and interact with other neurons (Thielscher et al., 2011; Esser et al., 2005). Thus, a cascade of neural activity is activated that can either remain localized at the site of stimulation or can spread by activating projection neurons to areas such as the spinal cord, cerebellum, thalamus and other areas of the cortex.

The intensity of a TMS pulse is often expressed relative to the intensity required to elicit a minimal MEP (usually in a hand muscle) from M1: this is known as motor threshold, qualified as resting (assessed at rest) or active (assessed during liminal voluntary contraction of about 5% max) (Fig. 1). At this intensity, TMS elicits action potentials in cortical pyramidal neurons that project out of the cortex to the spinal cord (the corticospinal axons) (Hallet, 2007). Here they activate excitatory synaptic connections to spinal motoneurons, and if the discharge is sufficiently powerful, it will fire the motoneurons and result in a MEP.

**A simple model of MEP production.** To understand the MEP fully it is important to note the many different ways in which these processes can happen. In the simplest scenario, the TMS pulse activates large-diameter corticomotoneuronal neurons. These represent the fastest conducting component of the corticospinal tract and have monosynaptic excitatory connections to spinal motoneurons. When a TMS pulse is administered with the commonly used posterior–anterior (PA) current direction, at least two central nervous system synapses must be activated to evoke a MEP: the synapse in the cortex that discharges the pyramidal neuron, and the corticomotoneuronal synapse in the spinal cord (Di Lazzaro & Rothwell, 2014) (Fig. 2). When both are activated sufficiently to discharge their respective postsynaptic neurons, they produce the earliest motor unit discharges in the MEP. Each synapse has its own threshold, which depends on the number of presynaptic terminals that are activated and the excitability of the post-synaptic neuron (Rossini et al., 2015). In effect, there is a threshold in the cortex and a threshold in the spinal cord. Similarly, the cortical and spinal synapses have their own input–output relationship, so the amplitude of the evoked MEP also depends on separate cortical and spinal mechanisms.

If we stick with this simple disynaptic pathway (there is another synapse at the neuromuscular junction, but since this is obligatory and rarely fails, we can ignore it), then what does the amplitude of the MEP represent? Even if all the spinal motoneurons were activated simultaneously, the motor units in the muscle would fire at different times because peripheral motoneurons have a range of conduction velocities (Thomas et al., 1959). Thus, their negative and positive peaks would occur at different times and thus may smoothen and wash out each other to a longer sum potential. We can get an idea of the magnitude of this effect in a hand muscle by comparing the maximum compound muscle action potential (CMAP) evoked by stimulation of the peripheral nerve at Erb's point to the CMAP evoked from stimulation at the wrist (Mallik & Weir, 2005). The CMAP at Erb has an amplitude of about

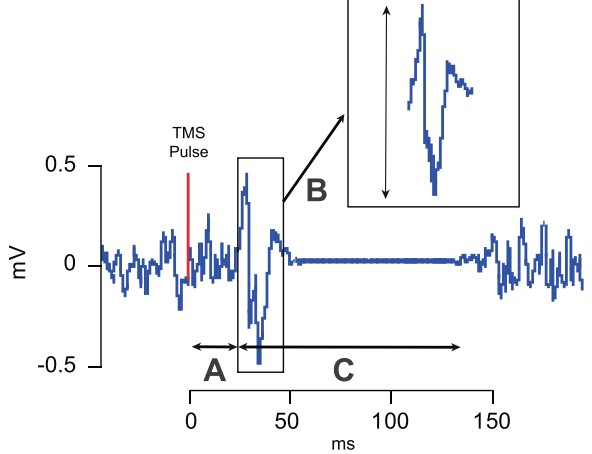

**Figure 1. Features of the MEP**
The motor *threshold* reflects the minimum intensity that elicits a small MEP in 50% of trials and can be probed at rest and during tonic muscle contraction. Motor thresholds are thought to rely on the excitability of cortico-cortical axons since voltage-gated sodium channel-blocking drugs increase thresholds (Ziemann, 2013). It is important to note that motor thresholds vary tremendously across individuals and do not represent a static measure. For instance, differences in scalp-to-cortex distance and genetic factors may explain a lot of the variance in the motor threshold between individuals. *A*, *MEP latency* provides information about the conduction time for the neural responses triggered by TMS to reach the targeted muscle, including the time for excitation of cortical neurons, conduction of the pyramidal tract and summation of descending volleys at the spinal level, and conduction time of peripheral motor neurons. The latency, therefore, can be influenced by both the state of the cortical and spinal motor neuron pool and certain stimulation parameters. For instance, MEP latencies are shortened with voluntary contraction as this action reactivates spinal motoneurons and in turn lowers their firing threshold. *B*, *MEP size* can be measured either by measuring peak-to-peak amplitude or the area under the curve of the rectified MEP. With either measure, it is possible to test TMS recruitment curves that establish the input–output relationship between increasing TMS intensity and resulting MEP size. While measuring MEP size as peak-to-peak amplitude is more commonly used, this metric is only valid when there is no occurrence of polyphasic oscillations. For example, MEPs elicited for non-hand muscles tend to be more polyphasic (Groppa et al., 2012), as well as those recorded from various patent populations, including patients with the amyotrophic lateral disease (Kohara et al., 1999), myoclonus dystonia (Van Der Salm et al., 2009), multiple sclerosis (Kukowski, 1993) and stroke (Brum et al., 2016). *C*, *silent period* represents a period of reduced electrical activity that follows the MEP when elicited during voluntary contraction. The duration depends on the stimulus intensity and is influenced by both intracortical and spinal mechanisms. The initial portion has been suggested to be due to a contribution from spinal mechanisms involving changes in motoneuron excitability and recurrent inhibition (Fuhr et al., 1991). The latter part has been linked to cortical inhibition mediated by $GABA_B$ (Ziemann et al., 2004).

75% that of the wrist CMAP (e.g. Arunachalam et al., 2003). This represents effects only in the peripheral nerve. If we add in additional dispersion from conduction in the corticospinal tract plus synaptic relays, then the effects of dispersion on MEP amplitude should be considerable.

**Towards a real model of MEP production: multiple volleys.** The model of a single disynaptic, rapidly conducting pathway is highly simplified: unlike electrical stimulation of a peripheral nerve where all axons are activated simultaneously, activation of corticospinal axons is trans-synaptic and is generated by a cascade of neural connections in the cortex (Di Lazzaro & Rothwell, 2014).

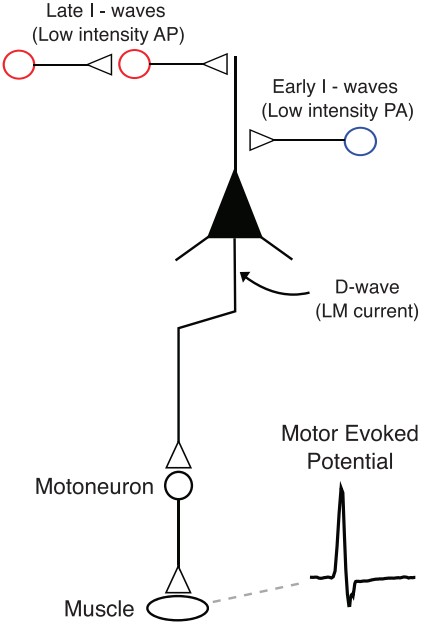

**Figure 2. Elements that contribute to the MEP signal**
Applying a suprathreshold transcranial magnetic stimulation (TMS) pulse over the primary motor cortex (M1) leads to a trans-synaptic activation of pyramidal cells, evoking descending volleys in the pyramidal axons projecting on spinal motoneurons (i.e. pyramidal tract). Motoneuron activation then leads to a contraction of the targeted muscle evoking a motor-evoked potential (MEP). The effect of cortical volleys on spinal cord circuitry can be evaluated by conditioning the Hoffman (H)-reflex (e.g. via peripheral nerve stimulation) with TMS. It is also critical to consider the direction of the TMS current, as this can change the cortical elements recruited with stimulation. When TMS is applied with lateral-to-medial (LM) currents, pyramidal cells are activated directly (D-wave). However, other current directions recruit pyramidal cells indirectly (I-waves) through the activation of excitatory interneurons. For instance, low-intensity TMS applied with a poster-to-anterior (PA) current preferentially recruits early I-waves (blue circle), whereas anterior-to-posterior currents tend to recruit late arriving excitatory inputs (late I-waves; red circle) that are different from ones sensitive to PA. Of note, MEPs can be elicited in hand, arm, leg and face muscles, albeit with different stimulation parameters. For instance, hand muscle MEPs are triggered most effectively with PA currents with relatively low stimulus intensities, whereas leg and face MEPs are more easily produced with lateral–medial currents. The threshold to elicit hand MEPs is low primarily due to hand muscles having the most extensive cortical representation that is located most superficially to the hemispherical surface. Importantly, while hand, arm and leg responses involve the corticospinal pathway, face muscle responses involve pyramidal neurons synapsing with motoneurons in the brainstem via the corticobulbar pathway.

Experimental approaches in rodents and non-human primates recording single corticospinal cell responses to TMS have shown that a single pulse of TMS evokes a cascade of high-frequency activity in the stimulated region (Mueller et al., 2014; Li et al., 2017). In humans, a suprathreshold TMS pulse evokes (i) multiple descending corticospinal volleys (as shown by invasive epidural electrodes placed over the high cervical cord) (Di Lazzaro et al., 2004) and (ii) multiple peaks of increased firing in the post-stimulus time histograms (recorded from single motor units of the targeted muscle) (Day et al., 1989). The multiple descending volleys occur at intervals of about 1.5 ms. These are known as the D-wave, which is followed by multiple I-waves that are numbered in their order of appearance (Patton & Amassian, 1954; Di Lazzaro & Rothwell, 2014). Typically, to stimulate the hand or arm area of M1, a figure-of-eight coil is oriented perpendicular to the central sulcus at an angle of 45 deg to the interaural line. If the TMS pulse is monophasic, the stimulation threshold is lowest when the coil induces a PA current in the brain. In this position, a threshold stimulus evokes the first, I1-wave, sometimes accompanied by the I2 and I3 waves (Di Lazzaro et al., 2004). As the intensity is increased, the waves increase in amplitude and number, and at some point, an earlier wave, the D-wave, preceding the I1-wave is recruited (Day et al., 1987). It should be noted that each pyramidal neuron rarely discharges in each I-wave: it may fire only once, at any of the peaks of I-wave periodicity.

The first I-wave results from monosynaptic inputs to corticospinal axons (Di Lazzaro et al., 2012; Fisher et al., 2002), whereas subsequent later I-waves likely result from polysynaptic inputs that include both inter- and intra-cortical connections to pyramidal tract neurons (PTNs). The D-wave reflects direct activation of the corticospinal tract axon, probably at the grey–white matter interface where the axon bends into the white matter (Di Lazzaro & Rothwell, 2014).

The underlying physiology of the I-waves is still unclear and could result from a combination of timed synaptic inputs and the repetitive firing properties of large pyramidal neurons (see Ziemann, 2020 for review). If synaptic input is the prime driver of I-wave periodicity, then the precise timing of the I-waves requires not only excitatory but also inhibitory inputs to sculpt neuronal excitation. The excitatory effect of excitatory synaptic input lasts for at least the duration of the rising phase of an excitatory postsynaptic potential, which is usually of the order of 2 ms. Since this is longer than the I-wave periodicity, it must be cut short by (probable feedforward) inhibition before the next wave of excitatory input arrives to produce the next I-wave. Pharmacological experiments have suggested that the likely neurotransmitters involved are glutamate and GABA (see Ziemann, 2020 for review).

Given that the I-waves (and D-wave) can continue for over 5 ms, there is considerable dispersion in the arrival of excitatory input at the spinal motoneuron and subsequent firing of motor units in muscle (Rossini et al., 2015). The consequence is that the MEP is always considerably smaller (peak-to-peak) than the CMAP and longer in duration. The number of volleys increases with TMS intensity and the MEP becomes increasingly polyphasic; indeed, some motor units may even fire twice in response to a single TMS pulse, once when the initial volleys arrive at the spinal cord and again when later volleys arrive (Day et al., 1987, 1989). Effects of dispersion vary from muscle to muscle, depending on the complexity of motor unit innervation (some muscles have more than one motor point) and conduction distance from the cortex.

**Towards a real model of MEP production: multiple pathways.** The monosynaptic corticomotoneuronal pathway is the most direct connection from M1 to muscle, but many other pathways are likely to be activated in addition. Corticospinal neurons synapse with many different types of neurons in the spinal cord so that di- and even polysynaptic pathways could be activated in addition to the monosynaptic connection. For example, there is clear evidence for transmission through proprio-spinal connections (Pierrot-Desilligny, 2002). Inhibitory spinal neurons mediating Ia reciprocal inhibition and Ib inhibition are also recruited, as well as neurons that modulate presynaptic inhibition (e.g. Kato et al., 2002).

Other descending tracts may be activated by TMS, such as the cortico-reticulospinal tract. Direct evidence for this exists in primates (Fisher et al., 2012) as well as supportive indirect evidence in humans (Maitland & Baker, 2021). Finally, we have no information on whether TMS activates small diameter slower-conducting corticospinal neurons, which represent more than 90% of the total fibres (Schüz & Braitenberg, 2002). If these are activated, then descending excitation will continue to reach the spinal cord for well over 10–20 ms. This may contribute to the late facilitation of H-reflexes observed in arm and leg muscles following a single TMS pulse to M1 (e.g. Wiegel et al., 2018).

**Summary of TMS-evoked MEPs.** Because TMS produces synaptic activity in the cortex, many potential excitatory and inhibitory pathways can participate in the production of the muscle MEP (Hallet, 2007). Since the efficiency of transmission at each synapse depends on both pre- and postsynaptic excitability, we might expect MEP amplitude to be influenced by a multitude of factors operating at the cortical, brainstem and spinal levels. Understanding changes in MEP amplitude in different experimental conditions or pathology requires that we understand which factor(s) are most likely to account for the results and devise experiments to exclude other factors. The idea that MEP amplitude can be used to quantify corticospinal excitability is true only in the very broadest sense that 'corticospinal' encompasses any possible set of synaptic connections and pathways that connect motor areas of the cortex to spinal motoneurons.

## Measuring MEPs: dispersion

The size of the MEP largely depends upon fluctuations in spinal motoneuron excitability and the distribution of synaptic activation across the motoneuronal pool. As discussed, the descending cortical volleys are capable of influencing motoneuron activation through two pathways: (i) monosynaptic connections to motoneurons (Lemon, 2008), and (ii) polysynaptic connections via projections to spinal interneurons that, in turn, synapse to motoneurons (Nielsen et al., 1993; Pierrot-Desilligny, 2002). Depending on these influences, some motoneurons may not fire or may even discharge multiple times, leading to a desynchronized discharge (Groppa et al., 2012). As a result, peak-to-peak measures of MEP amplitude are confounded by dispersion in the time at which excitatory input arrives at spinal motoneurons, with the result that the maximum MEP amplitude is considerably less than the maximum CMAP even when the latter is evoked from electrical stimulation of a proximal part of the nerve (e.g. Erb's point in the arm). Phase cancellation caused by dispersion can be accounted for with an elegant triple pulse stimulation technique that utilizes a collision method to resynchronize corticomotor excitation at the peripheral level and avoid the effects of dispersion (Magistris et al., 1998). Since this method is capable of depolarizing almost all spinal motoneurons and can account for their repeated discharges, the MEP amplitude recorded matches very closely to the maximally evoked CMAP in healthy individuals (MEP/CMAP ratio near 1). Although seldom used in a basic research setting (likely due to its difficulty to perform and discomfort), triple-pulse stimulation counteracts the variable amount of temporal desynchronization along the corticomotor pathway and helps explain an important source of MEP variability.

## Measuring MEPs: variability

One of the problems in using MEP amplitude as a measure of the response to TMS is trial-to-trial variability which is perhaps not surprising given the multiple factors that contribute to the response (Fig. 3). MEPs are sensitive to a complex combination of several ongoing neurophysiological processes at the time of stimulation, including the participant's mental state, sensory inputs received, state of cortical rhythms, phase cancellation of descending action potentials and the change in

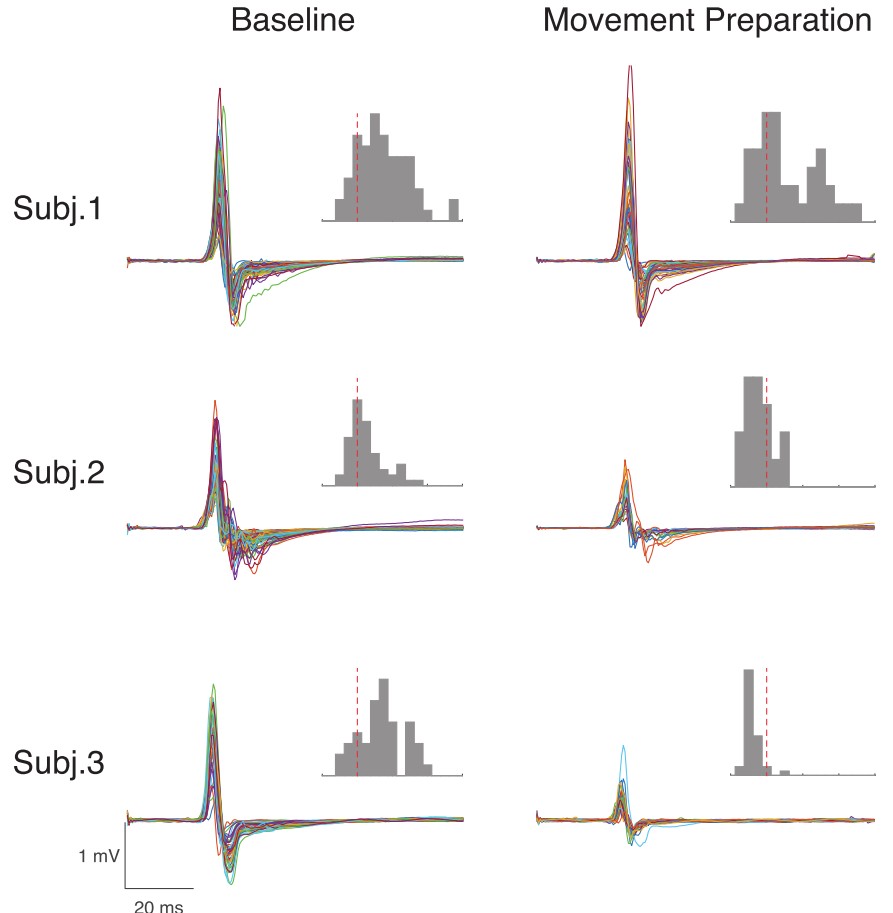

**Figure 3. Example of MEP amplitude variability within and across individuals**
MEP traces and amplitude histograms are provided for three participants across different experimental conditions. Recordings are from the FDI muscle. 'Baseline' refers to a condition where participants are waiting to receive a warning cue indicating that they need to get ready to react fast to a subsequent 'imperative' cue by flexing the index finger (warned reaction time task). In 'movement preparation', the TMS pulse was delivered approximately 200 ms before the average reaction time of the individuals in response to the imperative cues. The TMS intensity was set to obtain a 1 mV peak-to-peak amplitude during initial recordings performed while individuals were at rest. The dotted red lines on the histograms are aligned with a value of 1 mV on the *x*-axis. The distributions of MEP amplitudes in the histogram plots show that the response distribution and skewness can change between individuals and within individuals across different brain states. The data used for this figure were extracted from an original dataset (see Ibáñez, Hannah et al., 2020, https://doi.org/10.1093/cercor/bhz283).

synchronization of motor neuron discharges at the spinal level (Groppa et al., 2012; Bergmann, 2018; Siebner et al., 2022). Moreover, a variety of other biological and experimental factors are known to play a role in MEP variability, which has challenged the field to develop physiological and technical strategies to minimize their influence (see Table 1 for contributing factors). For example, a study using EEG-triggered TMS showed that MEPs are larger when evoked during the up-states than during down-states of slow oscillations in non-REM sleep (Bergmann et al., 2012), demonstrating that the state of the brain at the time of stimulation determines the overall response. Conversely, other strategies exploit measures of MEP variability to give an insight into processes such as motor preparation and neuroplasticity (Galea et al., 2013; Klein-Flugge et al., 2013; Goldsworthy et al., 2021).

When considering variability, a related problem is how best to quantify the MEP. For example, many early studies have estimated MEP amplitude by averaging the response of 10 trials, which can be problematic for various reasons. First, averaging the MEP response may not always be appropriate given that MEP amplitudes have a non-Gaussian distribution that can be skewed depending on the stimulation parameters selected (Goetz et al., 2014),

the state of the targeted muscle (rest or tonic activation) (Darling et al., 2006), age of the participant and patient population tested (example in multiple sclerosis – Britton et al., 1991). For example, MEPs recorded near threshold values tend to display right-skewed distributions that would likely overestimate MEP size when averaged, whereas ones recorded near saturation levels (e.g. high TMS intensity) display left-skewed distributions that likely underestimate MEP size (Goetz et al., 2014). Moreover, the first few MEP responses can be larger than subsequent MEPs (Schmidt et al., 2009). This increased variability due to these initial MEPs could be accounted for by removing the first trials, which may improve reliability. It is also important to note that MEP amplitudes are not time-invariant. The size and variability of responses change depending on the inter-pulse stimulus interval length (Julkunen et al., 2012), with evidence suggesting that intervals of 5 s or longer should be utilized as they increase the reliability of TMS measures (Pitkänen et al., 2017; Hassanzahraee et al., 2019). Even if these issues are accounted for, averaging from a small sample of responses can be potentially biased by outliers (Goetz et al., 2022); thus, considering the median response, as opposed to the mean, can be preferable as it is less distorted by

**Table 1. Experimental and biological factors contributing to MEP variability and solutions to overcome them**

| Factors of variability | Solutions to overcome/mitigate |
|---|---|
| **Internal** | |
| Gender, age and handedness | Appropriate sample size and design (e.g. gender match) |
| Genetic factors | Pre-screening for genetic factors (e.g. BDNF) |
| State-dependent factors | |
| Cortical oscillatory activity | Real-time EEG-triggered TMS tuned to specific oscillations; entrainment of oscillations (e.g. TMS during online tACS) |
| Attention | Maintain arousal (e.g. limit talking, keep eyes open, etc.) |
| Circadian rhythms | Perform experiments during the same time of day |
| Hyperexcitability | Removal of first 2–3 MEPs recorded; design experiments with matched predictability of TMS delivery |
| Muscle state (e.g. rest, activation) | Monitor consistent tonic muscle activation; set appropriate exclusion criteria |
| Anatomical factors | |
| Cortical thickness/white matter properties | Individualized E-field modelling |
| Interneuron recruitment | Apply different TMS current directions (e.g. PA *vs*. AP) |
| Synchronization of motoneuron discharges | Preform triple-pulse stimulation technique |
| **External** | |
| Coil positioning on scalp (e.g. tilt angle) | Use of neuronavigational systems |
| Coil orientation | Consider adjusting pulse widths for optimal PA and AP TMS |
| Stimulus floor/ceiling effect | Assess MEP amplitudes across various intensities (e.g. recruitment curves) |
| Stimulation parameters | Selecting intensity based on recruitment curves; applying appropriate inter-pulse intervals |

outliers. To account for the skewed MEP distribution, recent efforts have developed a statistical approach to calculate the number of trials to estimate single-subject MEP amplitudes based on subject variability (Ammann et al., 2020). Their results demonstrate that the minimum number of trials for estimating an individual MEP amplitude depends on the experimental condition and the amount of error considered acceptable by the experimenter.

Investigating the entire recruitment curve of MEPs at different stimulus intensities (input–output curves) provides better cortical excitability characterization than MEPs recorded at one set intensity (Kukke et al., 2014). The input–output curves reflect the gain in MEP amplitude with increasing stimulus intensity, which grows exponentially and eventually plateaus to saturation at high-intensity levels (e.g. sigmoidal function). In other words, increasing the intensity does not simply increase the number of recruited corticospinal fibres. It also influences the temporal dispersion of spikes propagating along the corticomotor pathways (Rossini et al., 2015). To fully understand stimulus–responses curves for a particular muscle, these curves can also be compared across different states, such as during consistent voluntary contraction. Voluntary contraction reduces temporal dispersion of the descending volleys, shortens MEP latency, steepens input–output curves, and facilitates corticospinal excitability relative to MEPs recorded at rest (Groppa et al., 2012). As such, some studies have argued that voluntary muscle contraction helps to stabilize cortical and spinal excitability and thus leads to decreased MEP variability (Darling et al., 2006; Kukke et al., 2014).

In support of this notion, Capaday (2021) recently developed a mathematical model based on basic neurophysiological properties that could explain this relationship and the sigmoid shape of the MEP input–output curve. In this model, $\alpha$-motoneuron discharge characteristics are considered as binary threshold units, in which the units are driven by noisy synaptic input currents with a Gaussian distribution. When the unit responds to noisy inputs, the discharge variance *versus* the response probability displays the inverted parabolic profile between MEP variance and MEP amplitude, a result supported by experimental data (Goetz et al., 2014; Capaday, 2021). Importantly their model shows that MEP variances increase with the level of motoneuron excitability independently of MEP amplitude. In other words, while the MEP amplitude is sensitive to changes in background muscle EMG, the model failed to demonstrate any significant correlation between MEP amplitude and different muscle activation levels (e.g. rest, 5% or 10% activity with respect to maximum voluntary contraction). One important implication of the model is that if motoneuron pool activity is controlled by having participants produce a constant muscle contraction, MEP variability can be attributed to corticospinal synaptic transmission and is

due to fluctuations of synaptic transmission at corticospinal terminals.

## Strategies to localize factors responsible for changes in MEPs

Since the TMS pulse recruits a variety of neuronal populations, one potential solution to reduce variability is to improve the selectivity of TMS pulses by adjusting stimulation parameters to target specific neuronal populations. To understand how this can be achieved, the next section will break down the elements that comprise the MEP and will introduce how specific TMS protocols can dissect distinct inputs to PTNs and spinal motoneurons.

**Recruitment of different I-waves by changing the TMS orientation, pulse width and waveform.** Although we have little direct information on the underlying neurophysiology of the circuits governing early and late I-wave generation, there is good evidence that they are linked to different subsets of physiological and behavioural plasticity (Hamada et al., 2014) and respond differently to various cortico-cortical afferent inputs (for a review, see Spampinato, 2020; Opie & Semmler, 2021); therefore, understanding how these circuits can be selectively targeted with TMS opens the opportunity to study distinct inputs to the M1.

The effect of changing the current direction and intensity of TMS is particularly clear in the hand area of M1, but similar principles are likely to occur at all cortical sites. In the hand area, coil orientation determines the probability of recruiting different D- or I-waves, and in consequence influences the onset latency of the MEP. For instance, currents applied in the lateral–medial direction are more likely to trigger D-waves and produce MEPs with shorter latencies (Di Lazzaro et al., 1998). Epidural recordings of the spinal cord have revealed that corticospinal volleys evoked by TMS show that PA currents preferentially recruit early I-waves, whereas anterior-posterior (AP) currents tend to recruit late I-waves. The differences of I-wave recruitment with different TMS current directions have also been demonstrated with single motor unit recordings (Day et al., 1989). The recruitment of later I-waves produces a series of excitatory postsynaptic potentials that temporally summate and discharge motoneurons at a longer latency than D-waves (Rossini et al., 2015). The consequence is that PA MEPs have a shorter latency than AP MEPs when recorded also with EMG (Di Lazzaro et al., 1998). Similar effects can be observed when TMS is used to facilitate the monosynaptic H-reflex: there is greater facilitation of the H-reflex when the first I-wave arrives at the spinal level after PA TMS compared with AP TMS (Niemann et al.,

2018). Overall, these studies indicate that MEPs produced with opposite TMS current directions reflect the activity of separate synaptic inputs to PTNs.

The ability of AP currents to recruit late I-waves varies between individuals (Hamada et al., 2013; Hordacre et al., 2017), mainly because AP stimulation produces less synchronized cortical activity (Di Lazzaro & Rothwell, 2014). Fortunately, recent work utilizing a controllable TMS device that can modify pulse width and waveform has shown that adjusting these parameters can help combat this difficulty. Standard monophasic TMS pulses compared to biphasic TMS are more likely to recruit distinct neuronal populations as they produce an electric field in one direction (Sommer et al., 2006). Modulating TMS pulse widths can also improve the selectivity of targeting specific neuronal populations based on the principle that one can activate axons of neurons that have different strength–duration properties (Mogyoros et al., 1996). For instance, AP currents with shorter pulses ($\sim$30 $\mu$s) evoke longer MEP latencies when compared to standard pulse durations ($\sim$100 $\mu$s) (Hannah & Rothwell, 2017), suggesting that manipulating pulse width duration alters the proportion of early and late synaptic inputs to corticospinal neurons (D'Ostilio et al., 2016; Hannah & Rothwell, 2017). Ultimately, this demonstrates that modifying parameters such as TMS pulse width and shape, along with the use of different current directions, can improve the efficacy of recruiting specific neuronal populations.

It is important to stress, however, that there is little information about which neural elements are targeted with TMS and how they relate to I-wave generation (Ugawa et al., 2020; Ziemann, 2020). Recent modelling work suggests that PA TMS activates a boundary region between caudal dorsal premotor cortex and anterior M1 located around the posterior lip of the precentral gyrus (Aberra et al., 2020). Corticospinal neurons that originate from this region rarely have monosynaptic connections to spinal motoneurons and do not have the fastest-conducting corticospinal axons (Siebner et al., 2022). It is thought that stimulation here activates cortico-cortical connections to caudal M1 where the majority of corticomotoneuronal, rapidly conducting fibres originate (Siebner et al., 2022; Dubbioso et al., 2021; Weise et al., 2020; Aberra et al., 2020). AP currents are thought to activate a spatially segregated premotor neural population that is more anterior in the precentral gyrus than the site recruited with PA stimulation (Aberra et al., 2020). It is likely that activation of corticocortical fibres with AP currents may subsequently activate neurons in caudal M1, presumably via a different pathway to that used by PA stimulation, to recruit a later set of I-waves. The implication is that: (i) differential effects of TMS current direction may originate upstream from neural populations within M1, and (ii) anatomical

inter-individual differences within the dorsal premotor cortex may contribute to between-subject MEP variability.

## Using paired-pulse TMS to probe specific neural populations

As noted above, MEPs provide information about the corticospinal connection in its entirety, a concept that subsumes a host of different anatomical pathways and synaptic relays. Applying an additional TMS pulse can sidestep some of these limitations and allow researchers to probe the excitability of specific populations of M1 interneurons and test the connectivity between M1 and brain areas projecting to it (Fig. 4*A*). In these protocols, an initial conditioning pulse is applied to a specific brain region (e.g. M1, premotor areas, cerebellum) before delivering a test stimulus over M1. Depending on its intensity and timing, the conditioning pulse can have either inhibitory or facilitatory effects upon M1 (Kujirai et al., 1993; Wagle-Shukla et al., 2009; Fong et al., 2021), estimated by changes in the test MEP amplitude. As a result, paired-pulse TMS can provide insights into the neurophysiological interaction between populations of cortical neurons.

Not only are these methods useful to identify mechanisms that differentiate healthy and patient populations, they can also give information about how particular pathways contribute to different behaviours. This is because all these interactions, like the MEP itself, are 'state-dependent' in that the observed effects depend on the excitability (or 'state') of the pathway at the time of testing (Bergmann, 2018). It is assumed that if a pathway is active in a task, then its excitability will differ from that at rest. It is commonly assumed that excitability will increase during active use. Postsynaptic neurons will be more depolarized and hence more likely to respond to pre-synaptic transmitter release (Silvanto & Pascual-Leone, 2008). However, there is always the possibility that the opposite may happen because ongoing synaptic activity increases the conductance of the postsynaptic neuron and reduces the amplitude of excitatory postsynaptic potentials (Paulus & Rothwell, 2016).

**Single-site (M1) paired-pulse TMS.** The most widely used method is short-interval intracortical inhibition (SICI), which employs a sub-motor threshold conditioning pulse to activate inhibitory circuits that primarily depress late I-waves and reduce MEP amplitudes when delivered 1–5 ms before the test stimulus (Kujirai et al., 1993; Fig. 4*A*). SICI is thought to depend on the activity of GABA$_A$ receptors since the administration of benzodiazepine, a positive allosteric modulator of GABA$_A$ receptor, increases the SICI response (Di Lazzaro et al., 2006; Ziemann, 2013). Facilitation of the MEP response

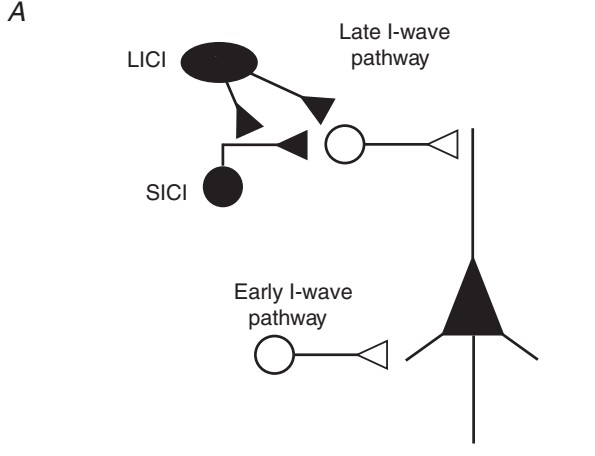

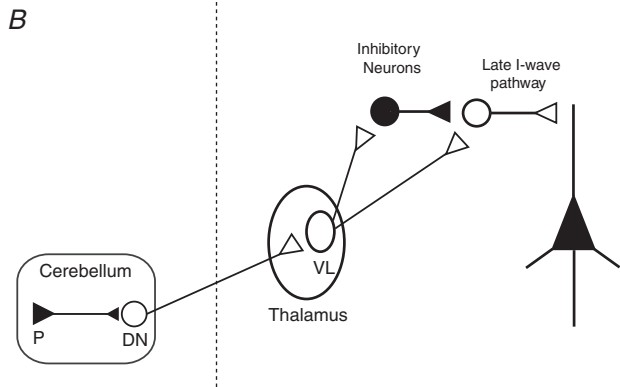

**Figure 4. Probing pathways within M1 and cortical-cortical connectivity with TMS**

*A*, Interregional interactions from both excitatory (open circles) and inhibitory (filled circles) neuronal populations can modulate the output of M1. For instance, the use of paired-pulse TMS over M1 can probe inhibitory processes like short- and long-interval intracortical inhibition (SICI and LICI), which have been shown to modulate late I-waves. Moreover, triple-pulse TMS protocols can be used to see how different neural populations interact with one another (e.g. LICI *vs*. SICI). Using this strategy, LICI has been shown to reduce the effect of SICI (Sanger et al., 2001), a phenomenon that has been suggested to occur by presynaptic GABA$_B$ receptor (LICI)-mediated inhibition (Sanger et al., 2001; McDonnell et al., 2006; Muller-Dalhaus et al., 2008). *B*, modulation of the MEP has also been found when a conditioning pulse is administered in a temporally specific manner to the cerebellum and parietal, premotor, or supplementary motor cortices. In the example above, one can use paired-pulse TMS to study the connectivity between the cerebellum and M1 via the cerebellar–thalamic tracts (Spampinato & Celnik, 2018), which have been found to interact with both excitatory and inhibitory interneurons (Daskalakis et al., 2004; Fong et al., 2021).

occurs at longer interstimulus intervals, particularly at slightly higher conditioning intensities, of between 10 and 15 ms (intracortical facilitation). Another type of facilitation, termed short-latency intracortical facilitation (SICF), can be observed when the test stimulus is instead followed by a conditioning stimulus at interstimulus time

intervals of 1.5, 2.9 and 4.5 ms (Ziemann et al., 1998) and at an intensity around or just under threshold. As these timings coincide with I-wave periodicity in epidural spinal cord recordings, the facilitation is thought to reflect the interactions between excitatory postsynaptic potentials generated by the two pulses within intracortical circuits that generate I-waves (Ziemann et al., 1998; Hanajima et al., 2002; Di Lazzaro et al., 2012). Long-latency intra-cortical inhibition (LICI) is another protocol that reduces the MEP response when two supra-threshold TMS pulses are given ~150 ms apart and is thought to be mediated by $GABA_B$ receptors (Werhahn et al., 1999; McDonnell et al., 2006). Beyond the involvement of different receptor sub-types for SICI and LICI, the suprathreshold stimulation of LICI will excite neurons that are spatially different from those in the SICI protocol, as the subthreshold conditioning pulse in SICI will stimulate more superficial parts of the precentral gyrus.

The important feature to note about all these inter-actions is the crucial role of timing and intensity. Subtle adjustments of each allow us to maximize our chances of observing the action of particular populations of neurons. Each method activates many different circuits; we just arrange our observation window to highlight one or other main factor. For example, the facilitation in intra-cortical facilitation is contaminated by continuing SICI and we observe facilitation only because it predominates at that latency (Wagle-Shukla et al., 2009). Similarly, both SICI and SICF will be activated during SICF, but stimulus timing focuses attention on the I-wave inter-actions (Peurala et al., 2008).

Another form of inhibition can be assessed by using peripheral nerve stimulation. Electrical or mechanical stimulation of a particular peripheral nerve produces afferent activity (sensory input) from the contralateral limb that reaches M1 through thalamo-coritical afferents or the somatosensory cortex (Hamada et al., 2012). The effect of this afferent input on the MEP amplitude depends on the time between electrical stimulation of the targeted nerve and a supra-threshold TMS pulse over M1 (Tokimura et al., 2000). For instance, MEP suppression is observed when the interstimulus time inter-val between nerve stimulation and M1 TMS is either short (20–25 ms, short-latency afferent inhibition; SAI) or long (200–1000 ms, long-latency afferent inhibition; LAI). Importantly, these effects occur only if the homo-topic stimulation of sensory input matches the location of the muscle targeted by TMS (Di Lazzaro & Rothwell, 2014). Pharmacological studies have shown that SAI and LAI both decrease with benzodiazepines lorazepam and diazepam, which involve $GABA_A$ $\alpha1$ receptors subtypes (Di Lazzaro et al., 2005; Turco et al., 2018). Beyond the interstimulus time interval differences between SAI and LAI protocols, SAI is known to suppress selectively late I-waves and not the first I-wave (Tokimura et al.,

2000) and is further modulated by cholinergic circuits. This is supported by clinical findings which have found decreased SAI in patients with cognitive deficits such as Alzheimer's disease (Di Lazzaro et al., 2002) and mild cognitive impairment (Nardone et al., 2012). On the other hand, abnormal LAI has been described in patients with sensorimotor impairment (Sailer et al., 2003; Morgante et al., 2017), but not in patients with cognitive deficits.

All these techniques rely on measuring changes in the MEP amplitude. In many cases, control experiments have been performed that localize the locus of the change to cortical sites and exclude possible changes in synapses in the spinal cord of the brainstem. However, one factor poses a technical challenge: the results are very variable (even more than the MEP itself), particularly between individuals. It is always tempting to assume that differences between individuals are due to differences in the excitability of the circuits being tested, particularly if they are from neurological populations: 'poor SICI indicates reduced GABA-ergic activity'. But this is not necessarily the case. For example, SICI cannot be observed in some 5–10% of healthy individuals (Du et al., 2014; Wassermann et al., 2002), even though, by definition, they are 'normal'. One possible explanation for this may be due to differences in the proportion of I-waves recruited by the test pulse. As previously discussed, the recruitment of late I-waves across individuals is variable, and this can lead to interindividual differences in response to conditioning protocols. Moreover, part of the variability is also likely due to SICI being mediated by a heterogeneous ensemble of inhibitory interneurons; however, modifying the pulse width and current direction may prove useful to target a subset of interneurons. In summary, it is important to note that the data collected from paired-pulse protocols reflect a mixture of both the composition of the test MEP and the sensitivity of the conditioning protocol being tested. Therefore, careful attention is needed when interpreting correlations between a specific physiological mechanism and behavioural measures, as well as when comparing responses between patient populations.

**Measuring cortico-cortical connectivity with MEPs.** TMS is also a useful tool to investigate afferent connections to M1 from other motor and non-motor areas of the cortex. The 'two-coil' (or 'twin coil') approach evaluates the effect of a conditioning TMS pulse applied over a brain region of interest on a MEP evoked by a test stimulus applied over M1 (Fig. 4*B*) (for review, see Reis et al., 2008). The twin coil TMS approach has allowed the field to investigate the physiological interactions of M1 with other brain regions, such as bilateral posterior parietal cortices, ventral and dorsal premotor cortices, supplementary motor area, and the cerebellum. Several studies have shown that small-diameter coils are generally needed for these

interactions, along with careful coil positioning for the cortical sites that are very close together (Civardi et al., 2001; Koch et al., 2008; Davare et al., 2009). The resulting impact on MEP amplitudes (e.g. facilitation or inhibition) depends on the intensity (Civardi et al., 2001), current direction (Zoghi et al., 2003; Spampinato et al., 2020), interstimulus time intervals (Koch et al., 2008; Davare et al., 2009), and the ongoing activity in the conditioned brain region (Davare et al., 2009; Ziluk et al., 2010).

As noted above these connections are state-dependent. For instance, the connectivity between one region and M1 may dramatically change before movement initiation, following mental training of a motor behaviour or as a result of learned behaviour (i.e. pre/post-interventional comparisons); therefore, many previous investigations have attempted to describe the association between a specific change in connectivity and behavioural modifications (Olivero et al., 2003; Koch et al., 2008; Davare et al., 2009; Hasan et al., 2013; Spampinato & Celnik, 2018). Importantly, if such a link is found, it should be stressed that this does not prove a causal relationship between physiological changes and behaviour (Bestmann & Krakauer, 2015). This relationship may only be achieved when using TMS to elicit 'virtual lesions' of a target cortical area to disrupt its function and associated behaviour (Allen et al., 2007). Since acquiring new behaviours involves the interaction of many upstream cortical regions that can influence the motor system, we stress that caution should be taken when linking specific inhibitory or excitatory processes to a particular behaviour. Therefore, careful attention to the experimental design details (e.g. control site, brain state) that control for confounding variables is needed when attempting to characterize the role of different intracortical interactions in motor behaviour.

The recent development of multi-locus TMS coil presents as a promising avenue for the future testing of cortico-cortical pathways. With multi-locus TMS, one can effectively stimulate multiple regions at a high temporal resolution, without repositioning the coil (Koponen et al., 2018). In particular, this approach may be useful for stimulating surrounding regions of M1 (e.g. premotor and supplementary motor areas), as it eliminates the errors associated with using a second coil (e.g. reliable positioning and targeting) and may also be used to dissociate premotor from M1 effects on MEP amplitudes.

## MEPs to study voluntary motor control

TMS provides an effective way to investigate changes in movement-related preparatory activity because it can be precisely timed with specific phases of movement, thus MEPs can provide a temporally precise and muscle-specific readout of the motor system before,

during, and after motor behaviour (Bestmann & Duque, 2016). However, although TMS can be useful to study corticospinal changes in the context of motor processing, several considerations need to be made when MEP changes are associated with specific motor states. One critical aspect in this regard is the fact that the main factor responsible for enhancing MEPs in the transition from rest to activity is likely to be increased excitability of spinal motor neurones, rather than increases in the descending corticospinal volleys (DiLazzaro et al., 1999). Indeed, the engagement of spinal interneurons, particularly Ia-inhibitory interneurons receiving direct afferent inputs, plays a major role in the coordination of agonist–antagonist muscles (Côté et al., 2018). Recordings from spinal interneurons in monkeys demonstrate that preparatory activity also occurs at the spinal cord (Prut & Fetz, 1999). Indirect evidence has also been obtained in human studies that demonstrated changes in the H-reflex during the warning period of reaction time tasks (Hasbrouq et al., 1999; Duque et al., 2010) implying that changes in spinal excitability could contribute to the MEP.

Another relevant consideration is the distinction between excitability and cortical activation. Since firing rates of cortical neurons can change markedly during motor processes, it is common to interpret MEP modulations during motor tasks as being primarily driven by changes in the levels of firing activity in the brain areas targeted with the TMS (Riehle & Requin, 1989). However, firing rate changes should not be confused with excitability. For example, before movement initiation, corticospinal neurons are known to increase their firing up to several hundreds of milliseconds before movements begin (Evarts, 1966; Godschalck et al., 1981), while MEP amplitudes have mainly been found to increase a few tens of milliseconds (MacKinnon & Rothwell, 2000; Ibáñez, Hannah et al., 2020) or hundred milliseconds before the onset of muscle activity (Nikolova et al., 2006; Chen et al., 1998; Cirillo et al., 2021). The reason for this discrepancy is that MEPs only provide a crude measure of net corticospinal excitability of the targeted brain areas. This measure reflects changes occurring at multiple scales ranging from the cellular level to large-scale changes (such as global inhibitory mechanisms when initiated actions need to be aborted, as discussed later on in this section). At the cellular level, for example, increases in synaptic activation in a given cell during certain motor tasks may inherently lead to changes (decays) in its membrane resistance (Paulus & Rothwell, 2016). This would imply that additional transmembrane currents generated by synaptic inputs would have a smaller effect on the neural discharge rate. In other words, an increase in membrane conduction during the intense activation of a neuron may reduce MEP responses more strongly than one expects. Below we discuss a few examples of experiments that examine states of movement preparation

in which there are no changes in ongoing EMG activity and in which control experiments have been performed to try to localize the changes to cortical rather than spinal circuits.

MEPs may change in amplitude when aborting or stopping already initiated actions in response to an external cue (e.g. in a stop-signal reaction time task). To successfully stop an action, it is thought that global inhibitory inputs to the motor system via the cortical-subthalamic nucleus hyper-direct pathway allow for a fast and generalized suppression of motor cortical outputs to the spinal cord (Aron et al., 2016; Rawji et al., 2020; Fig. 5*A*). This phenomenon, termed 'global suppression' (e.g. suppression of corticomotor excitability to the right and left M1 somatotopic representations), has been demonstrated in experiments that have utilized TMS in this context. MEP recordings obtained during the period after the stop signal display reduced cortico-spinal excitability affecting what is present not only in muscles involved in the action planned, but also in the surrounding 'task-irrelevant' muscles, such as in leg muscles when stopping a hand movement (Jana et al., 2020; Greenhouse et al., 2012). This apparent broad suppression of corticospinal excitability suggests that MEP measurement in this context reflects the action of subcortical non-selective inhibitory inputs directly inhibiting PTNs to block potential M1 commands to the periphery.

MEP amplitudes can also be modulated by focal inhibitory connections in M1 only affecting subsets of corticospinal connections. An example of this is the model of surround inhibition commonly used to explain how the brain selectively recruits specific effectors for a given task without residually activating neighbouring, unwanted effectors (Sohn & Hallett, 2004). Surround inhibition is thought to be mediated by the indirect pathway of the basal ganglia which, controlled by frontal regions like the right inferior frontal cortex and pre-supplementary motor area (Aron et al., 2007), exerts an inhibitory influence on cortical motor outputs to muscles that are not required for a given action (Aron & Poldrack, 2006). In this context, surround inhibition has been suggested as a cortical mechanism to counteract the general increase in spinal excitability that occurs during movement initiation (Beck & Hallet, 2011). During action initiation with a specific effector (e.g. the first dorsal interosseous), MEPs recorded from a non-involved effector (e.g. the abductor digiti minimi) are suppressed, while they increase for the muscle producing the selected movement (Beck & Hallet, 2010; Ibáñez, Hannah et al., 2020). Such divergence in MEP changes observed in task-related and task-irrelevant muscles suggests a mode of tuned inhibitory control of motor cortical outputs supporting selective muscle activation (Fig. 5*B*). Previous work has suggested that local intracortical inhibitory interneurons may provide

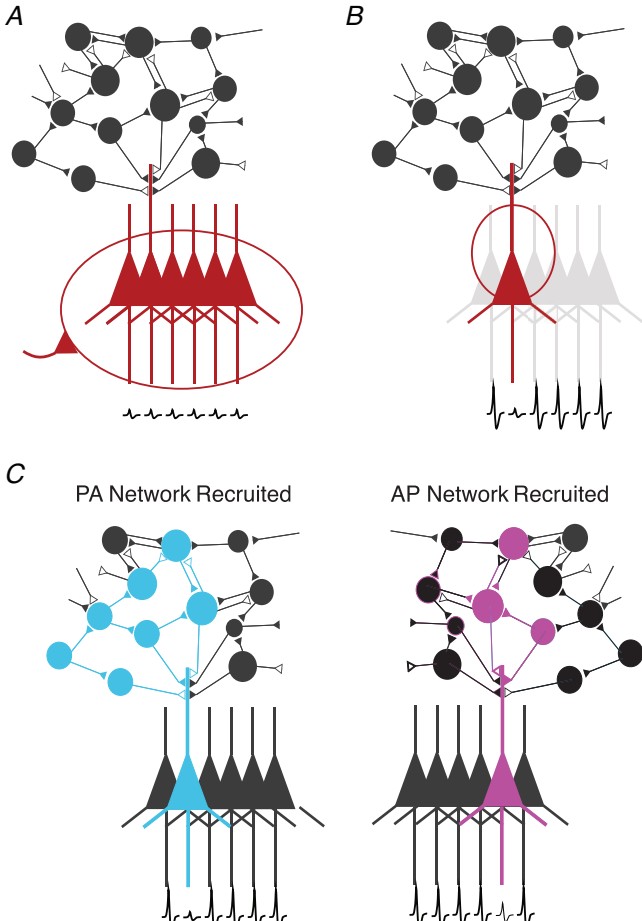

**Figure 5. Different cortical mechanisms may influence MEP amplitudes in the context of movement preparation and execution**

Filled triangles represent inhibitory inputs and open triangles represent excitatory inputs. PTN depicted in colours represents cells that are targeted in each scenario. *A*, global inhibition of motor cortical outputs that is task-unspecific (e.g. in the context of a stop-signal reaction time task). *B*, selective inhibition of circuits projecting to specific PTN muscle representations. This occurs in the scenario where selective activation of hand muscles is intended and is accompanied by a 'surround inhibition' effect suppressing cortical outputs to neighbouring, non-activated muscles. *C*, when TMS is given with different current directions, changes in MEP amplitudes may result in less effective depolarization of intracortical neurons due to network properties.

the neural substrate to achieve surround inhibition (Beck & Hallet, 2011).

Finally, MEP amplitudes can also reflect excitability changes in specific local cortical inputs to cortico-spinal neurons. This has been shown during periods of movement preparation in warned reaction time tasks (where a warning cue is given to allow subjects to prepare a movement before the presentation of the imperative cue to move). These tasks provide a useful tool to probe cortico-spinal excitability selectively during states of movement

preparation and initiation. Using this approach, several studies have shown a gradual decrease of MEP amplitudes in the muscles to be moved during the delay period between the warning and imperative cues (Hasbrouq et al., 1999; Duque et al., 2017; Ibáñez, Spampinato et al., 2020). Interestingly, this paradoxical reduction of corticospinal excitability affecting cells projecting to the soon-to-be-moved muscles is only observed for a subset of inputs to corticospinal neurons (those sensitive to AP currents) (Hannah et al., 2018). This level of selectivity in the modulation of the excitability of different pathways to the corticospinal tract may be interpreted in different ways. It may be achieved through feed-forward inhibitory mechanisms that can specifically target subsets of cortical neurons and pathways. However, this could require a high complexity in the way in which control of different M1 circuits is exerted to produce movements. Alternatively, selective excitability reduction may not represent an inhibitory input to the corticospinal system but an inherent property of the state and dynamics of the pathways and circuits that are targeted by the TMS when this is given with different orientations (Fig. 5C) (Kaufman et al., 2014; Ibáñez, Hannah et al., 2020).

To conclude this section, TMS can be used to track net excitability changes in action-specific localized and action-specific M1 circuits during motor processing. By designing adequately controlled experimental protocols, it is possible to measure excitability changes in specific cortical pathways associated with changes in M1 and its inputs at different spatio-temporal scales. This information combined with population activity measured with neuroimaging has advanced our understanding of the cortical neural processes during motor control thanks to the complementarity of these different 'windows' into the brain. Future research combining TMS with invasive recordings will be critical for understanding how cellular, synaptic and neural population changes contribute to MEP amplitude changes in the context of movement. Advances in this direction will maximize the information obtained from MEPs regarding how the brain generates and controls movements.

### Interpreting MEPs in pathological conditions

The interpretation of MEPs in disease poses specific problems beyond those mentioned in healthy brains. There are several reasons for this, the most notable being that similar changes in MEP features can result from different pathological processes. This section briefly reviews how MEPs, obtained with single or paired-pulse TMS, are affected by distinct neurodegenerative diseases.

As mentioned, MEP amplitude is the end-point of many synaptic relays between M1 and spinal motoneurons. Pathologies can affect any or all of these pathways. For example, amyotrophic lateral sclerosis (ALS) is a form of motor neuron disease involving progressive degeneration of both PTNs and $\alpha$-motoneurons, the latter eventually leading to damage of motor axons, muscle denervation and atrophy. Due to this widespread impairment, changes in MEP amplitude can be challenging to interpret, as they might not be reflective of processes occurring in M1; this can lead to a selection bias in clinical studies (Calancie et al., 2019). A partial solution to this problem is represented by the exclusion of subjects where the maximum MEP amplitude does not reach a certain value (Menon et al., 2015). Another possibility is to calculate the ratio between MEP and CMAP amplitudes. The latter represents an indirect measure of the residual spinal motoneuronal pool; thus, the MEP/CMAP ratio could give information about cortical excitability by factoring out the bias represented by motor unit degeneration (Weber et al., 2000). A further method to at least partially avoid the bias represented by spinal motor neuron degeneration would be to use triple pulse stimulation as a way to increase the MEP/CMAP ratio and provide more focused information on M1 integrity (Wang et al., 2019).

Even if the interpretation of the MEP itself poses problems, other methods that use the MEP can still give information about degenerative processes affecting M1. For example, several studies suggested that SICI is reduced in ALS (Vucic et al., 2008; Van den Bos et al., 2018), pointing to a breakdown of inhibitory neurotransmission involving GABA$_A$ receptors (Kujirai et al., 1993). This abnormality can be partially restored by riluzole, a drug whose mechanisms of action include blockage of tetrodotoxin-sensitive sodium channels and inhibition of glutamate NMDA receptors (Stefan et al., 2001; Geevasinga et al., 2016). Although SICI can give information beyond that provided by unconditioned MEP in the context of corticospinal tract damage, there are some caveats to its interpretation. One issue is that SICI decreases when the amplitude of the test MEP is small (Garry & Thomson, 2009), which is possible in ALS patients, due to the degeneration of motor units. Thus, the ability to elicit reliable test MEPs in patients is an important factor to consider in clinical studies. A second caveat is that, since SICI acts by suppressing I-waves (Di Lazzaro et al., 2012), a hypersynchronized and/or hyperexcitable state of excitatory M1 interneurons can result in a spurious decrease in SICI. This may occur in ALS, where increased effectiveness in summation of I-waves has been suggested in one study by using the short intracortical facilitation (SICF) paradigm (Van den Bos et al., 2018). In the same work, a correlation between SICI and SICF was found, further supporting the notion that abnormal I-wave facilitation can reduce SICI.

Another interesting question is whether the MEP can also be useful to study diseases where M1 is not specifically involved. Several studies have addressed the

information yielded by MEPs in Alzheimer's disease (AD), which is the most common neurodegenerative disorder affecting cognition. The pathology of AD is complex and a thorough description is beyond the scope of the present work, but in essence it is commonly thought that cortical damage particularly affects higher-order cortical association areas, while sparing primary sensory and motor cortices (Scheltens et al., 2016). While results of TMS investigations in AD patients have been mostly conflicting, some studies have shown a degree of loss of inhibition, mostly consisting of decreased SICI (Pierantozzi et al., 2004; Hoeppner et al., 2012), and decreased SAI, a putative marker of cholinergic neurotransmission (Mimura et al., 2021). In light of the pathological anatomy, it is difficult to interpret this result. One possibility is that the SICI findings are related to subtle and sometimes overlooked motor symptoms in AD. For example, patients can exhibit gait impairment and parkinsonian signs on examination (Schirinzi et al., 2018); Additionally, cortical myoclonus, which is associated with decreased SICI (Hanajima et al., 1998; Rocchi et al., 2019), is a common feature in late stages of the disease (Chen et al., 1991). However, some authors have hinted at a more informative role of SICI in these patients. Resting motor thresholds have been found to correlate with cognitive impairment assessed by the mini-mental state examination (Alagona et al., 2001; Khedr et al., 2011); this would suggest that M1 excitability reflects, to an extent, the global degenerative process occurring in AD. Overall, to further explore the relationship between MEP and neurodegeneration in diseases presenting with widespread brain cell loss, it would be advisable to couple TMS with structural investigations able to estimate the extent of cortical damage.

Another interesting issue concerns the MEPs in neurological diseases where the pathological process is mostly absent in M1 but present in cortical areas which are part of the network subserving control of voluntary movement. As such, these areas are directly or indirectly interconnected with M1 and can affect its function even in the absence of intrinsic structural damage. One example is Parkinson's disease (PD), whose pathological hallmark in early stages is represented by the loss of dopaminergic neurons in the substantia nigra pars compacta (Dickson, 2012). Although cortical damage can be present in PD, this occurs in late disease stages and often spares M1 (Dickson, 2012). Thus, changes in M1 excitability in PD are likely due to the alteration of input fibres connecting to M1. Most studies have found increased M1 excitability in PD at rest, reflected by a steeper input–output relationship on MEPs (Bologna et al., 2018), decreased SICI (Ridding et al., 1995; Kojovic et al., 2015; Bologna et al., 2018) and increased SICF (Ni et al., 2013). Voluntary contraction of the muscle targeted with TMS significantly reduces the input–output relationship in PD when compared to healthy subjects, further suggesting that the excitability of the motor system in these patients is abnormal (Valls-Solé et al., 1994). While these changes are thought to be most prominent in the hemisphere contralateral to the most affected body side (Spagnolo et al., 2013; (Kojovic et al., 2012), recent work with a large sample size of patients has also demonstrated that SICI is abnormal in the less affected hemisphere (Ammann et al., 2022). These alterations have been interpreted as compensatory mechanisms that facilitate voluntary movement where symptoms are more severe and are probably caused by decreased excitatory input from the basal ganglia to M1, rather than by intrinsic damage to the latter. If so, therapies aiming to restore basal ganglia physiology in PD should also affect M1. Indeed, several studies have confirmed this by showing that both deep brain stimulation of the subthalamic nucleus (Cunic et al., 2002) and administration of levodopa and apomorphine increase SICI, as well as other measures of M1 inhibition (Pierantozzi et al., 2002; Casula et al., 2017). However, the same caveat mentioned for ALS applies in the context of PD: since abnormally increased SICF has been reported (Shirota et al., 2019), decreased SICI should be interpreted with caution, including the possibility that it derives, at least partly, from increased excitability of excitatory interneurons.

In summary, MEP amplitude can give valuable insight into M1 excitability in selected patient populations, although some caveats should be kept in mind. These are mostly due to intrinsic limitations of the MEP, such as the fact that it depends on spinal cord physiology, as well as cortical, and that M1 dysfunction is not necessarily representative of global cortical impairment in some neurological diseases. As explained in the next sections, coupling TMS with techniques other than EMG may help to overcome some of these issues by expanding the number of possible readouts of TMS effects.

## Isolating information on M1 physiology using additional measures

**Measurement of twitch force recording.** Due to phase cancellation mechanisms mentioned above, MEP amplitude might not fully reflect descending activity generated by corticospinal tract neurons; this does not occur with twitch force, which is not subject to phase cancellation. This divergence is particularly clear for high stimulation intensities, when MEP amplitudes reach a plateau, while twitch force continues to increase (Kiers et al., 1995). Due to this, the measurement of force associated with MEPs has been proposed as a more accurate means to assess output from the corticospinal tract. However measurements of force from one muscle can be contaminated by concurrent activation

of antagonist and synergist muscles, making changes in force tricky to interpret.

**TMS-EEG.** Despite the abundance of neurophysiological and behavioural studies conducted with TMS, our overall understanding of MEPs and the mechanisms that underlie their generation remains incomplete. Recent technological advancements, however, such as the integration of TMS with EEG, provide a direct way to measure the effect of brain stimulation on brain activity, which may help expand present knowledge on the mechanisms responsible for MEP generation (Rocchi et al., 2018).

While MEP modulation can be affected by changes at a subcortical level and is limited to the study of M1, TMS-evoked cortical potentials (TEPs) recorded with EEG have the potential to record data without the influence of non-cortical confounds (Taylor et al., 2008; Rocchi et al., 2020). Like MEP, TEPs are sensitive to stimulation intensity, current direction and brain state (Kähkönen et al., 2005; Casula et al., 2018); however, one advantage of the TEP is that it can be recorded from local and distant electrodes. In other words, a TMS pulse is capable of probing the propagation of cortical signals in time and space across brain regions (Massimini et al., 2009; Casula et al., 2020). This may allow researchers to assess how changes in brain state affect neural activity (i.e. due to behaviour-related state changes (Fong et al., 2021)), including effects on whole functional networks, and localization of sources responsible for these activity changes, including the MEP (Thut & Miniussi, 2009). Another advantage is that TEPs have high interindividual reproducibility when stimulation is given over both motor and premotor cortices (Lioumis et al., 2009; Kerwin et al., 2018) and low levels of individual variation across multiple sessions (Matamala et al., 2018; ter Braack et al., 2019). In the clinical setting, TMS-EEG has emerged as a powerful tool to characterize biomarkers of treatment and pathophysiology of brain disorders (Tremblay et al., 2019). For example, TMS-EEG provides a novel way to detect the neural correlates of stroke-induced motor deficits (Tscherpel et al., 2020), including the identification of cortical reorganization following stroke (Pellicciari et al., 2018; Casula et al., 2021) and patient response to treatment (Koch et al., 2019; Tscherpel et al., 2020).

It should be noted that combining TMS with EEG remains challenging. While state-of-the-art EEG amplifiers are capable of dealing with time-varying magnetic fields without saturation, there still exist some difficulties in stimulating at high intensities and over certain brain regions (Veniero et al., 2009). For instance, if TMS is applied over lateral areas, including M1, EMG activity caused by direct scalp muscle activation can contaminate the EEG signal, requiring special cleaning techniques (Hernandez-Pavon et al., 2012; Rogasch et al., 2014; Salo et al., 2020). TEP can also be contaminated by afferent activity generated by muscle contraction associated with MEPs, in the case of suprathreshold stimulation of M1 (Fecchio et al., 2017; Petrichella et al., 2017; Biabani et al., 2021) and by EEG responses evoked by sensory input, the latter being represented by the TMS click (Conde et al., 2019) by depolarization of scalp somatosensory fibres (Nikouline et al., 1999; Gordon et al., 2018; Rocchi et al., 2021) and possibly by the twitch of cranio-facial muscles directly activated by TMS. There is evidence, however, that appropriate masking (i.e. masking noise, ear-defenders) can suppress auditory responses in some experimental settings (Massimini et al., 2005; Rocchi et al., 2021; Leodori, De Bartolo et al., 2022). In addition, EEG responses due to direct activation of cutaneous afferent fibres by TMS are very small when somatosensory input is mimicked by electrical stimulation of the scalp (Rocchi et al., 2021) or even absent when the TMS pulse is considered (Gosseries et al., 2015; Sarasso et al., 2020). Even if responses caused by indirect brain activation due to sensory input are present, they are represented by stereotypical vertex potentials in the 100–200 ms range, irrespective of the stimulation site, compatible with saliency-related multimodal responses (Mouraux & Iannetti, 2009; Rocchi et al., 2021); these show marked topographical differences compared to TEPs, the latter being characterized by maximal activity at the stimulation site and by the larger amplitude of signals of <100 ms (Belardinelli et al., 2019; Rocchi et al., 2021; Mancuso et al., 2021; Rawji et al., 2021). Beyond these issues, there is no general agreement on the preprocessing pipeline for removing early TMS-locked artifacts (i.e. cranial muscle activation and voltage decay). In the field, there is a debate as to whether this may (Bertazzoli et al., 2021) or may not (Mancuso et al., 2021) make an impact on the final TEP, and thus careful selection of TMS-EEG pipelines is recommended until a standard protocol is determined. Finally, it is common for studies to deliver 80–120 TMS pulses to obtain a reliable TEP and thus use short (∼2 ms) inter-pulse intervals to reduce the length of experiments (Rocchi et al., 2021). The short inter-pulse intervals are potentially problematic as they can modulate MEP amplitudes (Groppa et al., 2012) and thus possibly affect the amplitude of TEPs. However, recent work using very short intervals (1.1–1.4 ms) found no changes in TEPs over the course of stimulation (Leodori, Rocchi et al., 2022), suggesting that short pulse intervals can be used to reduce the duration of TMS–EEG studies without the risk of inducing potential changes related to the short stimulation rate. Interestingly, recent work has developed novel software capable of real-time monitoring of the data quality of TEPs that can help overcome the issues addressed above (Casarotto et al., 2022). The real-time

readouts from this software can help facilitate future studies as TMS parameters can be optimized based on a direct functional readout from the stimulated brain area before data acquisition begins.

There are limitations to TEP interpretation that should be taken into account. As with all EEG readouts, TEPs represent the combined activity of many different populations of neurones that overlap in time and spatial extent while the MEP represents the output of a small cortical area to a few hundred spinal motoneurons. Thus the TEP and the MEP very likely give very different forms of information. This is probably why there is some debate over whether it is possible to observe any correlate of the descending activity responsible for the MEP in the TEP. Some studies found correlations between MEP amplitude and N15/P30 (Mäki & Ilmoniemi, 2010) or N100 (Paus et al., 2001) TEP components; other authors, however, have not replicated these results (Bender et al., 2005; Bonato et al., 2006; Van Der Werf & Paus, 2006; Rocchi et al., 2018). Part of this discrepancy might be due to difficulties in understanding the sources of these TEP components, which have been reported to vary across areas other than M1 (Komssi et al., 2002); thus, they might not be necessarily informative about the dynamics underlying MEP generation, which are thought to take place within M1 during the first few milliseconds after the TMS pulse (Di Lazzaro et al., 2004). The spatial specificity of TEPs might also benefit from the use of spatial filters, source localization or other methods to account for volume conduction (Leodori et al., 2019; Rogasch et al., 2020).

In terms of investigating the involvement of distinct brain networks in cognitive and motor processes, both the offline and the online use of TMS-EEG allow scientists with novel approaches to complement the MEP. Offline approaches are particularly useful in the context of learning, as the neurophysiological correlates of learning a specific task can be achieved by comparing TMS-EEG responses (e.g. cortical excitability, oscillatory activity changes, functional connectivity) before and following task learning. Using EEG readouts, such as the lateralized readiness potential, together with *post hoc* sorting of MEPs constitutes another strategy for offline coupling between TMS and EEG, useful to investigate the relationship between cortical motor output and excitability of distributed brain networks in the context of action preparation (Verleger et al., 2006, 2009).

On the other hand, online use of EEG with TMS is a fairly recent concept that allows one to investigate whether particular oscillations of synchronized brain activity can shape the output response of M1, which can be used to develop more precise stimulation protocols that are tailored to the individual's ongoing brain state (Hannah et al., 2016; Zrenner et al., 2020). In these studies, EEG is used to monitor the amplitude and phase of ongoing oscillations (i.e. brain state), thereby allowing one to deliver a TMS-pulse in a time- and phase-locked manner and assess the relationship between oscillatory activity and MEPs (Berger et al., 2014). For instance, the MEP appears to be modulated by both the amplitude and phase of alpha (Schaworonkow et al., 2019; Bergmann et al., 2019) and beta oscillations (Torrecillos et al., 2020), indicating that phases of neuronal post-synaptic potentials have a significant impact on cortico-spinal communication. While a recent study found that mu power was positively correlated with corticospinal output (Wischnewski et al., 2022), it should be noted that there are conflicting results in the literature, with at least two studies reporting no modulation of MEP amplitude according to the phase of M1 mu rhythm (Madsen et al., 2019; Karabanov et al., 2021). Reasons for this discrepancy might lie in methodological factors such as the pre-selection of subjects based on EEG amplitude and differences in interstimulus time intervals and spatial filters used to extract the EEG signal. Future work will need to clarify how sensitive and stable the EEG responses are to varying task conditions (i.e. functional states), in order to provide insights into the relationship between oscillations, behaviour and TMS effectiveness.

Finally, it is important to consider that TMS-EEG measures physiological phenomena that are very different from the MEP. In comparison with the MEP, the effective 'recording site' is very large (i.e. lack of pin-point accuracy of looking at the output to just one muscle); however, the advantage of TMS-EEG is that the response is not filtered through the spinal cord. This means that one can look directly at cortical activity and see how (i) activity spreads from M1 to other areas of the cortex, and not just to the spinal cord, (ii) cortical areas other than M1 respond to TMS, and (iii) TMS interacts with ongoing cortical activity (how EEG affects the response to TMS and now TMS affects the EEG).

## Summary

TMS is a powerful tool that can provide insights into the cortical changes that occur following motor tasks or damage to the nervous system. We have summarized both the capabilities and limitations of TMS that are important to recognize when attempting to disentangle sources that contribute to the physiological state-related changes in cortical excitability. Thus, developing sound interpretations and conclusions requires a thorough understanding of the various cortical circuits and intrinsic factors that can influence the MEP. The development of specific stimulation protocols, such as applying different stimulation parameters (i.e. pulse width, current direction, paired-pulse), can overcome some limitations by providing ways to dissect specific contributions to the

MEP. Future studies could apply TMS in conjunction with other approaches such as TMS-EEG, as this combination can explore the dynamical state of neuronal networks within the stimulated region and interconnected areas. Overall, we hope this article serves as a useful guideline for the scientific community to use M1 TMS in behavioural and clinical settings.

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

## Additional information

### Competing interests

None.

### Author contributions

All authors have read and approved the final version of this manuscript and agree to be accountable for all aspects of the work in ensuring that questions related to the accuracy or integrity of any part of the work are appropriately investigated and resolved. All persons designated as authors qualify for authorship, and all those who qualify for authorship are listed.

### Funding

UKRI/Medical Research Council (MRC): D.A.S., J.I.P, L. R., J.C.R., MR/P006671/1. J.I. acknowledges funding from the Ramón y Cajal program of the Spanish Ministry of Science and Innovation (RYC2021-031905-I), supported by the State Research Agency (SRA) and European Regional Development Fund (ERDF).

### Keywords

motor cortex, motor-evoked potentials, movement, transcranial magnetic stimulation

### Supporting information

Additional supporting information can be found online in the Supporting Information section at the end of the HTML view of the article. Supporting information files available:

**Peer Review History**

