## [Peer Review History · The Journal of Physiology]

Motor potentials evoked by transcranial magnetic stimulation: interpreting a simple measure of a complex system

Danny Adrian Spampinato, Jaime Ibanez Pereda, Lorenzo Rocchi, and John C Rothwell
DOI: 10.1113/JP281885

Corresponding author(s): Danny Spampinato (d.spampinato@ucl.ac.uk)

Review Timeline:

Submission Date:	30-Mar-2021
Editorial Decision:	28-Apr-2021
Resubmission Received:	31-Aug-2022
Editorial Decision:	06-Oct-2022
Revision Received:	04-Jan-2023
Editorial Decision:	26-Jan-2023
Revision Received:	30-Mar-2023
Accepted:	18-May-2023

Senior Editor: Laura Bennet

Reviewing Editor: Richard Carson

Transaction Report:

Dear Dr Spampinato,

Re: JP-TR-2021-279614 "Motor potentials evoked by transcranial magnetic stimulation: interpreting a simple measure of a complex system" by Danny Adrian Spampinato, Jaime Ibanez Pereda, Lorenzo Rocchi, and John C Rothwell

Thank you for submitting your manuscript to The Journal of Physiology. It has been assessed by a Reviewing Editor and by 2 Referees and the reports are copied below.

Please let your co-authors know of the following editorial decision as quickly as possible.

As you will see, in its current form, the manuscript is not acceptable for publication in The Journal of Physiology. In comments to me, the Reviewing Editor expressed interest in the potential of this study, but much work still needs to be done (and this may include new experiments) in order to satisfactorily address the concerns raised in the reports.

In view of this interest, I would like to offer you the opportunity to carry out all of the changes requested in full, and to resubmit a new manuscript using the "Submit Special Case Resubmission for JP-TR-2021-279614..." on your homepage.

We cannot, of course, guarantee ultimate acceptance at this stage as the revisions required are substantial. However, we encourage you to consider the requested changes and resubmit your work to us if you are able to complete or address all changes.

A new manuscript would be renumbered and redated, but the original referees would be consulted wherever possible. An additional referee's opinion could be sought, if the Reviewing Editor felt it necessary. A full response to each of the reports should be uploaded with a new version.

I hope that the points raised in the reports will be helpful to you.

Yours sincerely,

Ian D. Forsythe
Deputy Editor-in-Chief
The Journal of Physiology
<https://jp.msubmit.net>
<http://jp.physoc.org>
The Physiological Society
Hodgkin Huxley House
30 Farringdon Lane
London, EC1R 3AW
UK
<http://www.physoc.org>
<http://journals.physoc.org>

EDITOR COMMENTS

Reviewing Editor:

Two thorough and thoughtful reviews of the submission have been received. As will be apparent, the referees concurred in their view that the review was well conceived, and that there existed the potential to both summarise the state of current knowledge concerning the motor evoked potential, and to inspire further empirical study. They also express the view - which I share, that the present version of the manuscript fails to capitalise fully on this potential. It can be appreciated that in writing a review of this kind, it is challenging to present material in a way that both caters to a novice in the field, and serves to challenge the current wisdom of the expert. It may be that in seeking to meet both of these challenges, neither has yet been addressed adequately. As the reviews are detailed and comprehensive, I will not seek to duplicate their content. By way of summary, there appears to be scope for the analysis provided in the review, to be both more precise and more penetrating.

Senior Editor:

As you see the referees and the Editor have made detailed critique of your review. While strengths are recognised, there are some major issues that need to be resolved before your Review could be considered for publication. Although I have rejected the MS, I would really encourage the authors to read the comments and if you are prepared to conduct the extensive revisions required, I would be delighted to receive a revised MS. If you do revise, please also consider that the

abstract needs to contain more factual information; avoid phrases like "Overall, this review aims to inform the scientific community about ...", your readers want to know the facts and have a clear conclusion. Please also consider revising the figures (perhaps adding a third) so as to provide a background to the key concepts and show how your synthesis of the data leads to a new interpretations. I hope that you find these reviews helpful in developing your ideas into a really important synthesis of your research topic.

REFEREE COMMENTS

Referee #1:

The authors submitted a narrative review in which they discuss advances and limitations of the motor evoked potential elicited with TMS over M1 in research on human corticomotor physiology. The review first describes variables that contribute to trial-by-trial variability of the MEP. The first two sections take up most of the review and can to some extent be considered a reminder of what we already know about the MEP. They delineate advances and limitations of estimating changes in corticomotor excitability. The second section summarizes how the MEP can be interpreted in the context of motor control and neurological conditions. The third section points out how EEG-based approaches can complement the MEP measure (i.e., TMS-EEG or more relevant to the overall aim of the review EEG-TMS).

GENERAL COMMENTS

This narrative review is a "missed opportunity". It is neither an opinion paper that takes an informed stand on an important controversy nor a progress report / perspective paper that outlines recent advances in the field and their significance for future research. The review has the length of a detailed physiological review that could thoroughly scrutinize the literature, but the review rarely goes beyond a superficial descriptive level. Many sections of this review fail to match the scientific level that one would expect from a review published in J Physiol. This reviewer has difficulties to see how this review would benefit the community. It provides no conceptual insights that go significantly beyond already existing topical reviews. Moreover, important aspects are not covered and there is some bias towards the authors own previous work rather than trying to covering the existing literature more broadly. Some major shortcomings of this review are listed below:

(1) The authors repeatedly call for caution and attention when interpreting TMS/MEP studies, but they do not disclose how the experiment should be designed to do better. How can researchers dissociate spinal from cortical effects? How can they dissociate premotor from primary motor or subcortical effects on MEP amplitude/area? The authors express their hope, that their paper may serve as a guideline, but the review provides no or little guidance apart from repeated calls for caution and attention.

(2) It appears that the authors are mainly describing MEPs collected from hand muscles, but this is not stated anywhere. This is important, as there is a difference in the contribution from direct corticomotoneuronal projections versus di and polysynaptic connections when comparing MEPs from intrinsic hand muscles as opposed to (e.g.) intercostals or soleus.

(3) The authors present TMS-EEG as a tool to reveal information of the "neural elements" excited with TMS, but they do not discuss the limitation of the TEP such as peripheral, non-transcranial confounds (e.g. activation by TMS-evoked muscle twitch, activation by somatosensory co-stimulation). The need of efficient auditory masking should be mentioned. The authors also fail to alert the reader to the problem of functional localization (sensor versus source space). The EEG read-out is also spatially biased towards the superficial cortex in the gyral crown, being less sensitive to "deep" neural elements in the sulcal wall or fundus of cortical gyri. The coverage of TMS-EEG and EEG-TMS should focus mainly on studies that investigated EEG correlates associated with variations in MEP amplitudes. They should also cite recent work that was less successful in finding clear effects of mu-power or phase on variations in MEP amplitudes and discuss in more depth methodological challenges (e.g. dependence of results on the spatial filter, inter-stimulus interval).

(4) The authors repeatedly warn the reader that the MEP also depend on the spinal circuitry, but they do not cite literature that has investigated this issue. They also do not offer any input as to how one should perform control experiments that can differentiate spinal from supraspinal contributions to the MEP. When summarizing and interpreting MEP studies on voluntary motor control, the authors completely ignore the possibility of global or focal inhibition as well as facilitation at the spinal

level (also in figure 2).

(5) The modulatory/conditioning effects of somatosensory or auditory stimulation on MEP amplitude, but also the effect of important physiological states such as NREM or REM sleep on MEP amplitude should be covered.

(6) The authors discuss MEP alterations in three neurodegenerative diseases (ALS, AD and PD), but do not discuss its diagnostic or prognostic value. The reader is very briefly introduced to ALS and AD, but do not really learn anything about how the MEP can provide insights into the underlying pathology. Instead, the authors describe the MEP/Mmax ratio as useful tool when studying disorders of the peripheral nervous system, although this ratio is relevant for more or less all studies using the MEP as a read-out. The authors do not mention the latency and how this is influenced by various neurological conditions even though this is well-studied.

Other properties of the MEP are only shortly mentioned such as the qualitative changes in the MEP (polyphasic configuration) and how these MEP features are related to diseases of the brain or spinal cord. In many neurological conditions (stroke, MS, SCI etc) MEPs are often polyphasic due to disorganized arrival from demyelination, reorganization and rerouting. A good way of assessing this could be to relate TMS evoked superimposed twitches to MEP area and amplitude or to use the triple pulse technique.

SPECIFIC COMMENTS

The review entails many physiological imprecisions that are listed below together with other comments. Since the authors have not added page numbers, this reviewer lists the comments in serial order without stating the corresponding pages.

Abstract:

MEP is not a readout of brain excitability but reflects corticomotor excitability.

Main text-Introduction

The statement that TMS offers high spatial resolution needs to be qualified. This reviewer argues that fMRI has a higher spatial resolution than TMS and more importantly, does not suffer from a spatial bias in sensitivity towards the hemispheric surface.

Several statements in the introduction lack appropriate references.

The MEPs indeed result from the summation of multiple volleys descending in several corticospinal projections. However, it is a bit simplified to state that it is transmitted through the fastest descending pathways (also provide references for such a statement). It is incorrect to state that it is the fastest descending corticomuscular pathways, as it is a compound potential, lasting tens of millisecond, which is ample time for di- and polysynaptic inhibitory and excitatory contributions. These include both intra- and intersegmental spinal interneurons. See e.g. the work by Nielsen and coworkers summarized in Petersen et al., (2003, EBR) as well as the work by Pierrot-Deseilligny (e.g. review in Muscle & Nerve, 2003). It should also be noted that in the preactivated muscle a single descending volley evoked by low-intensity TMS may be sufficient to evoke a MEP.

Terminology in the introduction is rather vague:

Is the MEP a complex signal? Maybe the mechanisms determining the MEP are complex rather than the signal itself?

'..TMS activates input-..' Is input here a, anatomical term? It is imprecise. Why not say projections?

'..extract as much information..' A minor detail but it is an odd phrasing. Especially in light of the absence of a discussion of the MEP latency.

'..measured MEP changes..' Only changes or also differences? The use in cross sectional studies comparing healthy controls to patients are often based on differences not only changes.

'..overview of the neuronal circuits involved in the generation of MEPs..' The authors provide a rough overview of the speculated cortical circuits activated by a TMS pulse. Maybe 'neural' is a better word? Also, it is not an overview, when the vast body of evidence showing contribution of many spinal interneurons in the MEP is not considered.

'..motor processing..' here, motor preparation or planning may be more appropriate.

Interpreting MEP variability in healthy brains

Important mechanisms determining MEP amplitude are not explained in sufficient depth. One prominent example is the impact of temporal desynchronization on MEP variability:

MEP amplitude may vary from trial to trial, even if transsynaptic activation of corticospinal volleys are identical. This may be caused by a variable amount of temporal desynchronization along the corticomotor pathway, causing different amounts of phase cancellation / temporal summation of the SMAP. This source of variability can be counteracted by the triple-pulse technique.

This reviewer would like to suggest that 'suprathreshold pulse applied to M1' should be replaced by 'suprathreshold pulse applied to precentral gyrus' as axon terminals in the gyral crown (BA6, rostral BA4a) has the lowest threshold (Aberra et al., 2020).

The initial paragraphs lack references.

The descending corticospinal volleys activate not only the spinal motoneurons but also the spinal circuitry.

The excitability and activity of cortico-cortical / thalamo-cortical axon terminals as well as intracortical interneurons play a role, not only PTN and spinal circuitry.

What is meant by more efficient temporal and spatial summation? Is this comment referring to corticomotor conduction and its temporal dispersion? Increasing stimulation intensity will activate a wider cortical area and likely increase firing rate in those interneurons activated with lower intensity, it also increases the excitatory/inhibitory drive to PTN as indicated by SIC1, ICF and SICF recruitment curves. This increases the number of action potentials in each PTN as well as the number of activated neurons. This causes the increase recruitment of motoneurons.

What do the authors mean with the term 'synaptic-fibres' ?

The use of the word 'dictate' is inappropriate, as it does not have superior control over the response as compared to the other factors.

'..a complex combination..' is imprecise. 'interactions' would be more precise.

I do not think that this was demonstrated by Brasil-Neto and coworkers in that paper. They have suggestions to why they see increased CV off the hotspot, but they do not interrogate the mechanistic underpinnings and they only suggest something partly similar to what they are cited for here.

One could argue that state-control also could be considered an experimental factor. Again, this does not aid the reader in conducting a TMS experiment with low variability. Why not systematically list the causes of variability divided in internal and external factors with the former further subdivided in state-dependent factor and anatomical factors? Then follow up with how to overcome/mitigate this.

What is the optimal number of MEPs? Several studies argue that >20 is needed for a reproducible read-out and more for ICF, SICI, SAI etc. How is the information that (paraphrased) 'Several trial are needed but not too many because it takes too long' useful for the reader?

MEPs 'elicited with high intensities..' What are high intensities? (e.g. conventional 150% of AMT) and why are high intensities needed? Electrical current strength needs to be sufficiently high to induce trans-synaptic excitation of PTNs deep in the anterior wall of the central sulcus.

The Aberra paper was published in 2020. Also, it is misleading to intermix experimental findings (Aberra) with a review (Spampinato) in the same citation. I do not think that the reference to Dr. Spampinato previous review helps the reader in this instance.

The Hamada 2013 and Hordacre 2017 references are not listed in the reference list.

'it is important to stress that the ongoing activity of M1 neuronal populations will ultimately dictate the MEP response.' Again, the word dictate is misleading. Which neuronal population? How does the ongoing activity influence the MEP response? This statement is too vague. The important contribution of spinal neuronal populations is again neglected. The importance of intrinsic neuronal excitability is also not mentioned here.

I do not understand which message the paragraph on focality of TMS effects wants to convey? What is meant with more resolution? And how can TMS have a higher resolution than e.g. laminar VASO? I cannot make sense of this nor the last sentence.

The entire section on paired-pulse TMS is confusing.

It should be clearly stated that this part is about single-coil paired-pulse TMS.

The sub-threshold conditioning pulse in SICI and LICI paradigms are exciting spatially different cortical regions (more

superficial parts of precentral gyrus) compared to the suprathreshold test stimulus (rather than different neuronal circuits in the same cortical patch).

SICF and SAI/LAI are not introduced.

What is meant by global excitability? The sum of activated inhibitory and excitatory inputs?

H-MRS of GABA does not reflect 'global levels of GABA neurotransmitters in M1'.

MEPs to study voluntary motor control

The effect of coil orientation has been investigated, so this can be cited and explained.

Here it would make sense to cite Zoghi and also Di Lazzaro.

'...a two coil approach'. Here the authors could mention the possibilities with the relatively newly developed coil that allows multi-site stimulation (Koponen et al., 2018, Brain Stimul),

The phrase '.. pathway than can exert... tones..' is jargon. Are the authors referring to transient inhibition or excitation?

Why does the integrating role of M1 make M1 an ideal target? This rather complicates the physiological interpretation of the MEP as stand-alone read-out.

The authors cite the review by Reis et al., but that does not aid the reader. Please cite the original sources and then complement with 'see Reis et al., 2008 for discussion' but only if the review supplements the original findings.

The authors repeat their claim that TMS has high spatial resolution. If the authors insist that TMS has a high spatial resolution compared to other neuroimaging modalities (fMRI, PET, MEG) or non-invasive stimulation techniques (TFUS?), they need to explain why and cite the relevant literature.

DiLazzaro 1999 is not listed in the reference list.

Ibanez 2020 is it Ibanez a or b?

The authors fail to acknowledge premovement excitability changes within the spinal circuitry see Prut & Fetz (1999, Nature) and Duque et al., (2010 J Neurosci). The corticospinal signal to dampen antagonist activity preceding agonist EMG onset (see figure in Griffin & Strick, 2020) would be expected to impact the spinal circuitry due to the divergence of the corticospinal system and the behavioral flexibility and many roles of spinal interneurons.

The authors need to clarify what is meant by global M1 suppression? Suppression of corticomotor excitability in the right and left hand/face/leg areas?

'..frontal cortical regions..'? This statement needs to be anatomically more specific. Also, as evident from the Griffin & Strick paper cited by the authors (and a plethora of other papers) inhibition of antagonist or non-active muscles has a prominent spinal component.

'Proactive'. Would 'feed-forward inhibition' be a more correct term?

It is not solely 'M1 excitability' that is reflected in the MEP amplitude. It is the state of excitability of inputs to M1, the cortico-cortical interneurons within M1, the PTN of M1, the spinal circuitry and the motoneuronal pool that determines its amplitude. The authors fluctuate a lot between structures they argue are responsible for the MEP amplitude.

'lower motoneurons'. Alpha motoneurons is a more specific term.

Interpreting MEP in pathological conditions

Here the triple pulse technique is highly relevant and should be mentioned. Alterations in corticomotor threshold and altered gain function (at rest and during tonic contraction) should be mentioned as relevant metrics.

Widespread impairment in ALS: Why is excluding patients a partial solution to this? Triple pulse stimulation may offer a solution here.

The Mmax is also influenced by skin preparations and electrode placement.

'Impaired' SICI. Maybe 'reduction in SICI' ?

In AD, reduced SAI is a relevant MEP marker indicating cholinergic deficit.

Narrowing down the MEP interpretations by combining them with complementary measurements

Here twitch force recordings should be mentioned as they are not subjected to phase cancellations.

The authors fail to make a convincing case how TEPs can narrow down MEP interpretations. This reviewer would argue that TEPs rather broaden the interpretations as the two readouts are very poorly related. It is unclear from the paragraph, how the TEP read-out contributes to our understanding of the MEP. Rather than referring to general review papers, the authors need to focus on TEP studies that directly relate MEP read-outs with TEP read-outs that have been (source) localized to the precentral gyrus.

The relationship between precentral TEP and MEP is a matter of debate, in part due to the contamination from peripheral activation. The authors are referred to the recent paper by Biabani et al (2021) in J Physiol. Here the authors need to

discuss the drawbacks of the TEP. It is heavily influenced by off-target effects that are difficult to mask and impossible to avoid. (see e.g. Conde et al. 2018, Neuroimage).

A way to relate EEG to MEPs is to use EEG to inform the timing of precentral TMS based on the expression of precentral oscillatory brain activity. Here, the literature is not as clear as the authors try to indicate (see Madsen et al. 2019, Brain Stim, Karabanov et al. 2021, Brain Stim). Based on a balanced view on the existing literature, one may conclude that the power or phase of pericentral cortical oscillations may account for a subtle fraction of trial-to-trial variability of the MEP amplitude, but this depends critically on methodological choices in terms of spatial filtering, inter-stimulus interval and number of stimuli.

Another option that should be mentioned is to apply post-hoc trial sorting of MEP trials according to ERP readouts (see Verleger et al. 2006, Exp Brain Res. and Eur J Neurosci. 2009).

Referee #2:

Summary

In the current manuscript, Spampinato and colleagues review the use of motor evoked potentials (MEPs) to study excitability of the corticospinal system. The authors emphasise that the MEP is a complex measure and that interpreting changes in MEP amplitude requires careful interpretation. The topic of the review is welcome considering how widely used the MEP is, and how often narrow interpretations of changes in MEPs are adopted. In general, the review is well written, however I do have some suggestions which I think will help to further refine some of the arguments raised by the author. In particular, I was expecting a stronger focus on the MEP itself - how is it measured, what does it look like, and how do changes in output from the motor cortex, spinal excitability, temporal dispersion etc impact MEP size and shape? I've detailed some of my suggestions below. I leave it up to the authors whether they choose to adopt them, however I do think they will further improve an already excellent review.

Major Comments

1. Given the title, I was expecting to read a more detailed account of the MEP itself, what is typically measured, and what can impact those measurements. Instead, there is a lot of discussion on basic physiology of TMS and motor output in terms of D-waves and I-waves, etc. These sections are really nice, and well written, but I found it a little unclear as to how changes to these outputs would impact the MEP. Furthermore, while most of the review seems to focus on the amplitude of the MEP, less space is given to other measures such as the threshold for evoking an MEP and the latency of the MEP. This is just a suggestion, but the authors might like to consider restructuring the first section to detail what an MEP is, how it is measured and what it typically looks like (possibly with an accompanying figure focusing specifically on the MEP and what is typically measured). They could then detail what is typically measured from an MEP (motor threshold, latency, peak-to-peak amplitude, area under the curve) and what is known to impact each of these measures. For example, the presence or absence of D-waves can alter the MEP latency, which changes as stimulation intensity is increased; or differences in scalp-to-cortex distance explains a lot of the variance in motor threshold between individuals however this measure is also sensitive to changes membrane potentials governed by sodium-gated ion channels as shown in pharmacological work (just as two examples). By detailing what can impact the properties of the MEP earlier in the manuscript, this will help to contextualise the two practical examples explained in the second half of the review (e.g., changes with motor control and in neurological disorders).

2. I'd suggest expanding the description and importance of desynchronised excitation of motor units in determining the size and shape of the MEP. I'd pay particular attention to the article by Magistris et al 1998, Brain as an example of how phase cancellation can impact MEP amplitudes. This is a crucial concept for understanding the size and shape of the MEP, especially when considering pathological impacts on the MEP (e.g., from demyelination). It might be helpful to compare the MEP size and shape to that of the compound muscle action potential (a good example is in Groppa et al 2012, Clin Neurophysiology). This is alluded to later in the review, but could be introduced earlier. Furthermore, it is important to fully explain what a compound muscle action potential is, and how it is generated/measured.

3. Pharmacological studies have taught us a lot about the different cortical mechanisms that MEPs are sensitive to, however this body of literature is barely discussed. I suggest including a more thorough description of how MEPs are altered by different pharmacological agents. Given that MEP amplitudes are altered by GABAergic drugs, this might help better contextualise the section discussing how MEPs are altered during movement preparation.

4. I was a little puzzled by the paragraph on MEP variability. Statements like 'One of the main challenges is dealing with relatively large trial-by-trial MEP variability' suggest that MEP variability simply reflects noise. I think the complexities of MEP variability are under-appreciated within the field and perhaps warrant a slightly more nuanced discussion, especially considering the title of the review. For example, the approach of taking the average MEP as a summary statistic ignores the fact that variability in MEP amplitude within an individual is rarely normally distributed, which raises the question as to whether this widely used approach is appropriate. Furthermore, the characteristics of MEP variability change with different stimulation intensities (from right skewed to left skewed and can be bimodal at some intensities; Goetz et al 2014 Brain Stimulation), with different states (e.g., at rest vs during a muscle contraction) and with different pathological disorders (example in MS - Britton et al 1991). Also, MEP amplitudes change over time and with different interstimulus intervals (e.g., Julkunen et al 2012 Brain Stimulation). All of these characteristics raise a lot of questions about how MEPs are measured and whether we miss a lot of critical information by simply taking the mean of the MEP at one stimulus intensity as opposed to more accurately characterising MEP variability. I think expanding the discussion a little will help to highlight the challenges of how best to measure the MEP given that it is variable, and challenge the field to think a bit more about what can we learn from this variability.

5. The paragraph on how changing coil angle impacts motor output would benefit from some more detail. PA vs AP changes the I-wave composition - how does this impact MEPs? AP is thought to stimulate premotor terminals - what about PA stimulation? What is the current consensus on where this stimulates?

6. In the TMS-EEG section, I suggest also discussing some of the limitations of this method (e.g., the recent debates regarding how much of the TEP represents sensory input with certain experimental arrangements - Gordon et al 2018, Conde et al 2019, Biabani et al 2019, Rocchi et al 2020 etc.).

Minor comments

1. Introduction: 'Finally, we discuss how the combination of TMS with electroencephalography (EEG) may represent as a possible strategy to more thoroughly study cortical components of the MEP.' I would suggest rewording this sentence. If I understand, I think what the authors are trying to convey here is that TEPs may represent an alternative measure for indexing cortical excitability similar to the MEP, but without the confound of spinal excitability. TEPs aren't really the cortical components of MEPs - they are sensitive to different spatial scales than MEPs (which focus on the output of a very small minority of corticospinal neurons, which is pointed out later in the review). But both may be sensitive to changes in cortical excitability.

2. Basic principles of TMS: 'minimal or no discomfort' - this is highly dependent on where the stimulation is applied. In some regions there can be some discomfort due to the stimulation of face/scalp muscles or nerves. I'd suggest rewording to reflect this.

3. 'Theoretically, this implies that tweaking the pulse parameters is one way to reduce the variability of MEP responses across individuals; however, it is important to stress that the ongoing activity of M1 neuronal populations will ultimately dictate the MEP response.' Is there any data to back this up? If not, I'd suggest removing as is overly speculative.

4. 'Finally, it is important to note that since the MEP represents the output of a small population of the entire pyramidal tract output, it is possible to make statements about the focality of effects in a way that has more resolution than fMRI or EEG. If one is able to conclude that the changes in MEP are due to changes in M1, the immediate secondary conclusion is that the changes in M1 are caused by changes in its input, therefore allowing one to infer changes happening in other, connected, parts of the motor circuits.' I found this paragraph confusing and I'd suggest rephrasing. I'm a little unclear what is meant by

'focality' and 'resolution' in this context. Are the authors suggesting that, due to the input-output nature of MEPs and the specificity to one muscle that a more causal inference is possible? How are changes located to M1 (presumably by ruling out spinal mechanisms)? Note the reliance on motor output is both a strength and weakness - it does allow a higher degree of specificity, but is highly limited to motor systems.

5. Figure 2B: The cartoon depicting surround inhibition could be a little more accurate. It's not clear that activation of this neuron suppresses other connected neurons (or vice versa - are the closed or open triangles representing inhibition)?

ADDITIONAL FORMATTING REQUIREMENTS:

-Please provide a legend to accompany the Abstract figure.

-Your MS must include a complete "Additional information section" with the following 4 headings and content:

Competing Interests: A statement regarding competing interests. If there are no competing interests, a statement to this effect must be included. All authors should disclose any conflict of interest in accordance with journal policy.

Author contributions: Each author should take responsibility for a particular section of the study and have contributed to writing the paper. Acquisition of funding, administrative support or the collection of data alone does not justify authorship; these contributions to the study should be listed in the Acknowledgements. Additional information such as 'X and Y have contributed equally to this work' may be added as a footnote on the title page.

It must be stated that all authors approved the final version of the manuscript and that all persons designated as authors qualify for authorship, and all those who qualify for authorship are listed.

Funding: Authors must indicate all sources of funding, including grant numbers. If authors have not received funding, this must be stated.

It is the responsibility of authors funded by RCUK to adhere to their policy regarding funding sources and underlying research material. The policy requires funding information to be included within the acknowledgement section of a paper. Guidance on how to acknowledge funding information is provided by the Research Information Network. The policy also requires all research papers, if applicable, to include a statement on how any underlying research materials, such as data, samples or models, can be accessed. However, the policy does not require that the data must be made open. If there are considered to be good or compelling reasons to protect access to the data, for example commercial confidentiality or legitimate sensitivities around data derived from potentially identifiable human participants, these should be included in the statement.

Acknowledgements: Acknowledgements should be the minimum consistent with courtesy. The wording of acknowledgements of scientific assistance or advice must have been seen and approved by the persons concerned. This section should not include details of funding.

-Author profile(s) must be uploaded via the submission form. Authors should submit a short biography (no more than 100 words for one author or 150 words in total for two authors) and a portrait photograph of the two leading authors on the paper. These should be uploaded, clearly labelled, with the manuscript submission. Any standard image format for the photograph is acceptable, but the resolution should be at least 300 dpi and preferably more. A group photograph of all authors is also acceptable, providing the biography for the whole group does not exceed 150 words.

Dear Reviewers,

Thank you for the positive reviews and insightful feedback on our review article. By addressing the concerns step-by-step brought up by each reviewer, we believe the quality of our manuscript has noticeably improved. Below we provide responses to the concerns raised by the reviewers (in red text):

Referee #1:

The authors submitted a narrative review in which they discuss advances and limitations of the motor evoked potential elicited with TMS over M1 in research on human corticomotor physiology. The review first describes variables that contribute to trial-by-trial variability of the MEP. The first two sections take up most of the review and can to some extent be considered a reminder of what we already know about the MEP. They delineate advances and limitations of estimating changes in corticomotor excitability. The second section summarizes how the MEP can be interpreted in the context of motor control and neurological conditions. The third section points out how EEG-based approaches can complement the MEP measure (i.e., TMS-EEG or more relevant to the overall aim of the review EEG-TMS).

GENERAL COMMENTS

This narrative review is a "missed opportunity". It is neither an opinion paper that takes an informed stand on an important controversy nor a progress report / perspective paper that outlines recent advances in the field and their significance for future research. The review has the length of a detailed physiological review that could thoroughly scrutinize the literature, but the review rarely goes beyond a superficial descriptive level. Many sections of this review fail to match the scientific level that one would expect from a review published in J Physiol. This reviewer has difficulties to see how this review would benefit the community. It provides no conceptual insights that go significantly beyond already existing topical reviews. Moreover, important aspects are not covered and there is some bias towards the authors own previous work rather than trying to covering the existing literature more broadly. Some major shortcomings of this review are listed below:

(1) The authors repeatedly call for caution and attention when interpreting TMS/MEP studies, but they do not disclose how the experiment should be designed to do better. How can researchers dissociate spinal from cortical effects? How can they dissociate premotor from primary motor or subcortical effects on MEP amplitude/area? The authors express their hope, that their paper may serve as a guideline, but the review provides no or little guidance apart from repeated calls for caution and attention.

The overall goal of this article is to provide a detailed and comprehensive view of MEPs, that is catered to individuals who might be less familiar with using TMS. We thought this article might be helpful to these individuals when they interpret the findings of studies that have utilized the MEP in different contexts (e.g. at rest and during/following behavior). To better achieve this goal, we have now included much more detailed sections that provide how researchers can dissociate the contribution of distinct areas (e.g. spinal, premotor, etc.) to the MEP from the primary motor cortex. Moreover, we have now provided a table that lists several factors that might contribute to variability in the MEP responses and how studies may be able

to control these factors (Table 1). Overall, we believe that the overall quality of the content has vastly improved, thanks to the helpful and constructive comments given by the reviewer.

(2) It appears that the authors are mainly describing MEPs collected from hand muscles, but this is not stated anywhere. This is important, as there is a difference in the contribution from direct corticomotoneuronal projections versus di and polysynaptic connections when comparing MEPs from intrinsic hand muscles as opposed to (e.g.) intercostals or soleus.

We thank the reviewer for bringing this important detail to our attention. The reviewer correctly points out our focus is on MEPs recorded in hand muscles; however, we agree and now mention the important differences of comparing MEPs recorded from face muscles compared to hand muscles, as the former will involve the corticobulbar tract and the latter involving the corticospinal tract. We have now added these details in the text of Figure 1.

(3) The authors present TMS-EEG as a tool to reveal information of the "neural elements" excited with TMS, but they do not discuss the limitation of the TEP such as peripheral, non-transcranial confounds (e.g. activation by TMS-evoked muscle twitch, activation by somatosensory co-stimulation). The need of efficient auditory masking should be mentioned. The authors also fail to alert the reader to the problem of functional localization (sensor versus source space). The EEG read-out is also spatially biased towards the superficial cortex in the gyral crown, being less sensitive to "deep" neural elements in the sulcal wall or fundus of cortical gyri. The coverage of TMS-EEG and EEG-TMS should focus mainly on studies that investigated EEG correlates associated with variations in MEP amplitudes. They should also cite recent work that was less successful in finding clear effects of mu-power or phase on variations in MEP amplitudes and discuss in more depth methodological challenges (e.g. dependence of results on the spatial filter, inter-stimulus interval).

We certainly agree with all the limitations of TEPs mentioned by the reviewer. Some of them are already mentioned in the text:

"...if TMS is applied over lateral areas, including M1, EMG activity caused by direct scalp muscle activation can contaminate the EEG signal, requiring special cleaning techniques (Hernandez-Pavon et al., 2012; Rogasch et al., 2014; Salo et al., 2020). TEP can also be contaminated by afferent activity generated by muscle contraction associated with MEPs, in case of suprathreshold stimulation of M1 (Fecchio et al., 2017; Petrichella et al., 2017), and by EEG responses evoked by sensory input, the latter being represented by the click and the activation of scalp somatosensory fibres associated with TMS (Nikouline et al., 1999; Rocchi et al., 2021)". (lines 588-592)

According to the reviewer's suggestion, we have added that direct activation of cranial muscle may represent a source of contamination as well (lines 592-593).

We also agree that spatial localization can be problematic in TMS-EEG studies, as spatial filters or other methods to account for volume conduction are seldom used, and that an intrinsic limitation of the EEG is that it mostly reflects activity in superficial neurons oriented perpendicularly to the scalp, being less sensitive to changes in membrane potentials of neural elements deeper and oriented tangentially to the scalp. This information has been added to the text (lines 606-609).

Lastly, we also agree that studies reporting negative results about the relationship between MEP amplitude and phase of cortical rhythms should be reported; references have been added to the text (lines 643-647).

(4) The authors repeatedly warn the reader that the MEP also depend on the spinal circuitry, but they do not cite literature that has investigated this issue. They also do not offer any input as to how one should perform control experiments that can differentiate spinal from supraspinal contributions to the MEP. When summarizing and interpreting MEP studies on voluntary motor control, the authors completely ignore the possibility of global or focal inhibition as well as facilitation at the spinal level (including Figure 2).

In the section titled “Basic Principles of MEPs elicited by M1 TMS”, we have now added a subsection (A Simple Model of MEP production) that describes in detail how the important role of spinal circuitry when interpreting MEPs. In addition, under the section “Measuring MEPs: Dispersion” we now include how the size and shape of the MEP importance of desynchronized excitation of motor units and how this can be overcome with the triple pulse technique. Finally, we have also added additional text to the section on “MEPs to study voluntary motor control”, in which we mention aspects of global and focal inhibition, as well as potential spinal contributions (lines 378-385). The figure the reviewer is referring to is now “Figure 4”, which is only considering cortical contributions of inhibition.

(5) The modulatory/conditioning effects of somatosensory or auditory stimulation on MEP amplitude, but also the effect of important physiological states such as NREM or REM sleep on MEP amplitude should be covered.

We thank the reviewer for this suggestion. We agree that it is very important to consider how different physiological states have an important influence on MEPs (and also TEPs). We have now expanded upon this topic in various sections throughout the manuscript (see Table 1 “Factors of Variability”; see sections “Interpreting MEP variability in healthy brains” (212-215), “Using Paired-Pulse TMS to probe specific neural populations” (lines 274-276) and “Isolating information on M1 physiology using additional measures” (lines 586-604).

(6) The authors discuss MEP alterations in three neurodegenerative diseases (ALS, AD and PD), but do not discuss its diagnostic or prognostic value. The reader is very briefly introduced to ALS and AD, but do not really learn anything about how the MEP can provide insights into the underlying pathology. Instead, the authors describe the MEP/Mmax ratio as useful tool when studying disorders of the peripheral nervous system, although this ratio is relevant for more or less all studies using the MEP as a read-out. The authors do not mention the latency and how this is influenced by various neurological conditions even though this is well-studied.

We understand the reviewers’ point, but the aim of the paragraph was not to give a thorough account of MEP changes in a large number of neurological conditions, nor to examine the role of the MEP from a clinical point of view. Our aim was rather to use a limited number of neurological conditions to exemplify how neurodegeneration might bias MEP interpretation. In this context, we included ALS, AD and PD as “models” to consider how damage to different parts of the nervous system (degeneration of spinal motor neurons, diffuse cortical atrophy, impairment in basal ganglia function) might bias the conclusions which might be reached by considering MEP amplitude. For this reason, and due to the fact that the manuscript is already quite long, we would prefer not to provide further clinical details or to discuss features of MEP (e.g., latency) that would not be relevant to the aim of the review.

Other properties of the MEP are only shortly mentioned such as the qualitative changes in the MEP (polyphasic configuration) and how these MEP features are related to diseases of the brain or spinal cord. In many neurological conditions (stroke, MS, SCI etc) MEPs are often polyphasic due to disorganized arrival from demyelination, reorganization and rerouting. A good way of assessing this could be to relate TMS evoked superimposed twitches to MEP area and amplitude or to use the triple pulse technique.

As requested by Reviewer 2, we have now added more details about the different features /properties of the MEPs in a newly added figure and its caption (see Figure 2). Within the figure caption, we mentioned how the size of the MEP can be measured either by examining the peak-to-peak amplitude or by looking at the MEP area and why quantifying the area under the curve is more appropriate in the presence of polyphasic MEPs (as seen in many patient populations).

SPECIFIC COMMENTS

The review entails many physiological imprecisions that are listed below together with other comments. Since the authors have not added page numbers, this reviewer lists the comments in serial order without stating the corresponding pages.

Abstract:

MEP is not a readout of brain excitability but reflects corticomotor excitability.

Fixed.

Main text-Introduction

The statement that TMS offers high spatial resolution needs to be qualified. This reviewer argues that fMRI has a higher spatial resolution than TMS and more importantly, does not suffer from a spatial bias in sensitivity towards the hemispheric surface.

Any text relating to TMS having high spatial resolution has now been removed.

Several statements in the introduction lack appropriate references.

Statements regarding information about TMS/MEPs are now appropriately referenced.

The MEPs indeed result from the summation of multiple volleys descending in several corticospinal projections. However, it is a bit simplified to state that it is transmitted through the fastest descending pathways (also provide references for such a statement). It is incorrect to state that it is the fastest descending corticomuscular pathways, as it is a compound potential, lasting tens of millisecond, which is ample time for di- and polysynaptic inhibitory and excitatory contributions. These include both intra- and intersegmental spinal interneurons. See e.g. the work by Nielsen and coworkers summarized in Petersen et

al.,(2003, EBR) as well as the work by Pierrot-Deseilligny (e.g. review in Muscle & Nerve, 2003). It should also be noted that in the preactivated muscle a single descending volley evoked by low-intensity TMS may be sufficient to evoke a MEP.

We thank the reviewer for bringing this oversimplified sentence to our attention. In the modified manuscript, the reviewer should view a new subsection titled “A simple model of MEP production” that has been constructed to more accurately introduce the several cortical and spinal components that are involved with MEPs. Moreover, we also elaborate on other possible pathways that are likely activated with TMS under the section “Towards a real model of MEP production: multiple pathways”, which references the important work suggested by the reviewer (lines 158-164).

Terminology in the introduction is rather vague:

Is the MEP a complex signal? Maybe the mechanisms determining the MEP are complex rather than the signal itself?

We thank the reviewer for pointing out this detail. We have now removed this phrase.

'..TMS activates input-..' Is input here a, anatomical term? It is imprecise. Why not say projections?

This phrase has now been removed from the updated manuscript.

'..extract as much information..' A minor detail but it is an odd phrasing. Especially in light of the absence of a discussion of the MEP latency.

We have now fixed this odd phrasing and have now added some details of MEP latency in the new figure (Fig 2), which also discusses different characteristics of the MEP (i.e. threshold, latency, size, etc).

'..measured MEP changes..' Only changes or also differences? The use in cross sectional studies comparing healthy controls to patients are often based on differences not only changes.

The word “changes” is now replaced with “differences”.

'..overview of the neuronal circuits involved in the generation of MEPs..' The authors provide a rough overview of the speculated cortical circuits activated by a TMS pulse. Maybe 'neural' is a better word? Also, it is not an overview, when the vast body of evidence showing contribution of many spinal interneurons in the MEP is not considered.

We thank the reviewer for their suggestion. Throughout the text, we now try to remain consistent with using the term “neural” in this context. As mentioned in our response to the reviewer's main point #4, we now discuss more details regarding spinal contributions to the MEP.

'..motor processing..' here, motor preparation or planning may be more appropriate.

Changed now from '*..motor processing..' to 'motor preparation'.*

Interpreting MEP variability in healthy brains

Important mechanisms determining MEP amplitude are not explained in sufficient depth. One prominent example is the impact of temporal desynchronization on MEP variability:

MEP amplitude may vary from trial to trial, even if transsynaptic activation of corticospinal volleys are identical. This may be caused by a variable amount of temporal desynchronization along the corticomotor pathway, causing different amounts of phase cancellation / temporal summation of the SMAP. This source of variability can be counteracted by the triple-pulse technique.

We thank the reviewer for this important suggestion. As detailed in our response to the reviewer's main concern #4, we have now added a new section "Measuring MEPs: dispersion" that address this issue and introduces triple-pulse TMS.

This reviewer would like to suggest that 'suprathreshold pulse applied to M1' should be replaced by 'suprathreshold pulse applied to precentral gyrus' as axon terminals in the gyral crown (BA6, rostral BA4a) has the lowest threshold (Aberra et al., 2020).

We thank the reviewer for this suggestion and have now switched the text regarding this issue in multiple phrases (see lines 252-253; 258-259).

The initial paragraphs lack references.

Fixed.

The descending corticospinal volleys activate not only the spinal motoneurons but also the spinal circuitry.

The text now reads: "Corticospinal neurons synapse with many different types of neurons in the spinal cord so that di- and even poly-synaptic pathways could be activated in addition to the monosynaptic connection. For example, there is clear evidence for transmission through propriospinal connections (Pierrot-Desilligny, 2002). Inhibitory spinal neurons mediating Ia reciprocal inhibition and Ib inhibition are also recruited, as well as neurons that modulate presynaptic inhibition (e.g. Kato et al., 2002)." (lines 159-164).

The excitability and activity of cortico-cortical / thalamo-cortical axon terminals as well as intracortical interneurons play a role, not only PTN and spinal circuitry.

We have now added the sentence "Since the efficiency of transmission at each synapse depends on both pre- and postsynaptic excitability, we might expect MEP amplitude to be influenced by a multitude of factors operating at the cortical, brainstem and spinal levels." (lines 174-176)

What is meant by more efficient temporal and spatial summation? Is this comment referring to corticomotor conduction and its temporal dispersion? Increasing stimulation intensity will activate a wider cortical area and likely increase firing rate in those interneurons activated with lower intensity, it also increases the excitatory/inhibitory drive to PTN as indicated by SIC1, ICF and SICF recruitment curves. This increases the number of action potentials in each PTN as well as the number of activated neurons. This causes the increase recruitment of motoneurons.

We apologize for the confusing text and have now removed this phrase from the manuscript.

What do the authors mean with the term 'synaptic-fibres' ?

This terminology is now removed from the text.

The use of the word 'dictate' is inappropriate, as it does not have superior control over the response as compared to the other factors.

Fixed.

'..a complex combination..' is imprecise. 'interactions' would be more precise.

The phrase is now changed to 'interactions'.

I do not think that this was demonstrated by Brasil-Neto and coworkers in that paper. They have suggestions to why they see increased CV off the hotspot, but they do not interrogate the mechanistic underpinnings and they only suggest something partly similar to what they are cited for here.

As we have dramatically restructured the section of "Interpreting MEP variability in healthy brains", this reference and the text associated with it have now been removed.

One could argue that state-control also could be considered an experimental factor. Again, this does not aid the reader in conducting a TMS experiment with low variability. Why not systematically list the causes of variability divided in internal and external factors with the former further subdivided in state-dependent factor and anatomical factors? Then follow up with how to overcome/mitigate this.

We thank the reviewer for their suggestions. The manuscript now includes 'Table 1' that lists out biological and external factors influencing MEPs, while also providing potential solutions to help limit these factors.

What is the optimal number of MEPs? Several studies argue that >20 is needed for a reproducible read-out and more for ICF, SICI, SAI etc. How is the information that (paraphrased) 'Several trial are needed but not too many because it takes too long' useful for the reader?

We agree with the reviewer that the original text was rather vague and not constructive for the reader. The entire paragraph regarding MEP variability has now been reconstructed, which now includes references that have acknowledged that a minimum of 20 pulses are needed for most measures (see lines 204-210).

MEPs 'elicited with high intensities..' What are high intensities? (e.g. conventional 150% of AMT) and why are high intensities needed? Electrical current strength needs to be sufficiently high to induce trans-synaptic excitation of PTNs deep in the anterior wall of the central sulcus.

We have now removed this phrase in the current version of the manuscript.

The Aberra paper was published in 2020. Also, it is misleading to intermix experimental findings (Aberra) with a review (Spampinato) in the same citation. I do not think that the reference to Dr. Spampinato previous review helps the reader in this instance.

The reference by Dr. Spampinato is now removed.

The Hamada 2013 and Hordacre 2017 references are not listed in the reference list.

These references are now added.

'it is important to stress that the ongoing activity of M1 neuronal populations will ultimately dictate the MEP response.' Again, the word dictate is misleading. Which neuronal population? How does the ongoing activity influence the MEP response? This statement is too vague. The important contribution of spinal neuronal populations is again neglected. The importance of intrinsic neuronal excitability is also not mentioned here.

This sentence has now been removed from the manuscript and as mentioned in our previous responses to the reviewer, we have now included information detailing spinal and neuronal contributions.

I do not understand which message the paragraph on focality of TMS effects wants to convey? What is meant with more resolution? And how can TMS have a higher resolution than e.g. laminar VASO? I cannot make sense of this nor the last sentence.

We thank the reviewer for making us aware that the paragraph and statements regarding TMS focality were not clear to follow. This paragraph has been entirely removed from the manuscript to make space for other important topics (e.g. spinal contributions, SAI/LAI, etc).

It should be clearly stated that this part is about single-coil paired-pulse TMS.

We apologize for the confusion regarding the section on paired-pulse TMS and we have now created new sub-headings that separate single-site TMS protocols (such as SICI/LICI/etc.) from dual-site TMS protocols that measure cortico-cortical connectivity.

The sub-threshold conditioning pulse in SICI and LICI paradigms are exciting spatially different cortical regions (more superficial parts of precentral gyrus) compared to the suprathreshold test stimulus (rather than different neuronal circuits in the same cortical patch).

We thank the reviewer for pointing out this mistake, we have now added the following the text: “Beyond the involvement of different receptor subtypes for SICI and LICI, the suprathreshold stimulation of LICI will excite neurons that are spatially different from those in the SICI protocol, as the subthreshold conditioning pulse in SICI will stimulate more superficial parts of the precentral gyrus.” (lines 300-303)

SICF and SAI/LAI are not introduced.

We thank the reviewer for the suggestion of adding these fundamental protocols. They are now included in the text.

What is meant by global excitability? The sum of activated inhibitory and excitatory inputs?

This sentence has now been removed.

H-MRS of GABA does not reflect 'global levels of GABA neurotransmitters in M1'.

We thank the reviewer for pointing this out. The new version of this text has now removed this inaccurate statement.

The effect of coil orientation has been investigated, so this can be cited and explained. Here it would make sense to cite Zoghi and also Di Lazzaro.

These references and information have now been added to the text.

'...a two coil approach'. Here the authors could mention the possibilities with the relatively newly developed coil that allows multi-site stimulation (Koponen et al., 2018, Brain Stimul),

We thank the reviewer for their suggestion. We believe this is a very exciting strategy that can be used in future studies to dissect the influences of surrounding regions on M1 (see lines 366-368).

The phrase '.. pathway than can exert... tones..' is jargon. Are the authors referring to transient inhibition or excitation?

This phrase has now been removed.

Why does the integrating role of M1 make M1 an ideal target? This rather complicates the physiological interpretation of the MEP as stand-alone read-out.

We agree that to some level this can make the MEP readout more complicated and to some level is a limitation to using dual-site TMS. We have now rephrased this sentence.

The authors cite the review by Reis et al., but that does not aid the reader. Please cite the original sources and then complement with 'see Reis et al., 2008 for discussion' but only if the review supplements the original findings.

Fixed.

MEPs to study voluntary motor control

The authors repeat their claim that TMS has high spatial resolution. If the authors insist that TMS has a high spatial resolution compared to other neuroimaging modalities (fMRI, PET, MEG) or non-invasive stimulation techniques (TFUS?), they need to explain why and cite the relevant literature.

We thank the reviewer for pointing out this oversimplified claim and have now removed the phrase that states TMS has high spatial resolution when compared to other techniques.

DiLazzaro 1999 is not listed in the reference list.

Fixed.

Ibanez 2020 is it Ibanez a or b?

Fixed- Ibanez 2020a

The authors fail to acknowledge premovement excitability changes within the spinal circuitry see Prut & Fetz (1999, Nature) and Duque et al., (2010 J Neurosci). The corticospinal signal to dampen antagonist activity preceding agonist EMG onset (see figure in Griffin & Strick, 2020) would be expected to impact the spinal circuitry due to the divergence of the corticospinal system and the behavioral flexibility and many roles of

spinal interneurons.

We thank the reviewer for their suggestion and for pointing out these two important studies showing changes in spinal circuitry in the context of movement preparation. The text now reads “Indeed, the engagement of spinal interneurons, particularly Ia-inhibitory interneurons receiving direct afferent inputs play a major role in the coordination of agonist-antagonist muscles (Côté et al., 2018). Recordings from spinal interneurons in monkeys demonstrate that preparatory activity also occurs at the spinal cord (Prut and Fetz 1999). Indirect evidence has also been shown in human studies that demonstrated changes in the H-reflex during the warning period of reaction time tasks (Hasbroug et al., 1999; Duque et al., 2010) implying that changes in spinal excitability could contribute to the MEP.” (lines 383-389)

The authors need to clarify what is meant by global M1 suppression? Suppression of corticomotor excitability in the right and left hand/face/leg areas?

We thank the reviewer for this point of clarification. The text now reads “To successfully stop an action, it is thought that global inhibitory inputs to the motor system via the cortical-subthalamic nucleus hyper-direct pathway allow for a fast and generalized suppression of motor cortical outputs to the spinal cord (Aron et al., 2016; Rawji et al., 2020; Fig 2a). This phenomenon, termed “global suppression” (e.g. suppression of corticomotor excitability to the right and left M1 somatotopic representations) has been demonstrated in experiments that have utilized TMS in this context.” (lines 408-413)

‘.frontal cortical regions..’? This statement needs to be anatomically more specific.

Fixed. The text now reads: “Surround inhibition is thought to be mediated by the indirect pathway of the basal ganglia which, controlled by frontal regions like the right inferior frontal cortex and pre-supplementary motor area (Aron et al., 2007), exerts an inhibitory influence on cortical motor outputs to muscles that are not required for a given action (Aron and Poldrack 2006).” (lines 423-427)

Also, as evident from the Griffin & Strick paper cited by the authors (and a plethora of other papers) inhibition of antagonist or non-active muscles has a prominent spinal component.

We thank the reviewer for pointing out the important role of spinal interneurons in the coordination of antagonist-agonist muscles. We have now added the phrase “Indeed, the engagement of spinal interneurons, particularly Ia-inhibitory interneurons receiving direct afferent inputs play a major role in the coordination of agonist-antagonist muscles (Côté et al., 2018).” (lines 383-385)

‘Proactive’. Would ‘feed-forward inhibition’ be a more correct term?

We thank the reviewer for suggesting a more appropriate term, we have now switched the term ‘Proactive’ to ‘feed-forward inhibition’.

It is not solely ‘M1 excitability’ that is reflected in the MEP amplitude. It is the state of excitability of inputs to M1, the cortico-cortical interneurons within M1, the PTN of M1, the spinal circuitry and the motoneuronal pool that determines its amplitude. The authors fluctuate a lot between structures they argue are responsible for the MEP amplitude.

We have now removed any phrases in this section that stated ‘M1 excitability is reflected in the MEP amplitude’.

‘lower motoneurons’. Alpha motoneurons is a more specific term.

Fixed.

Interpreting MEP in pathological conditions

Here the triple pulse technique is highly relevant and should be mentioned. Alterations in corticomotor threshold and altered gain function (at rest and during tonic contraction) should be mentioned as relevant metrics.

We thank the reviewer for this useful suggestion. Indeed, the TST, which has been introduced previously in the text in this new version of the manuscript, might represent a useful tool to increase the MEP/CMAP ratio and provide information about M1 in a less biased way than MEP alone. This information has been added to the text, together with a relevant reference (Wang et al., 2019) (lines 482-484). However, we would rather avoid mentioning motor threshold, as the variable itself is represented by the maximal stimulator output, and not MEP amplitude, the latter representing the focus of this review. We would also avoid mentioning alterations in MEP gain, as the point we are trying to make by discussing ALS is to elaborate on the bias represented by peripheral degeneration on MEP amplitude – something which would occur when measuring input/output MEP relationship as well.

Widespread impairment in ALS: Why is excluding patients a partial solution to this? Triple pulse stimulation may offer a solution here.

The authors' reasoning (Menon et al., 2015), which we concur with, is that MEP cannot reliably offer pathophysiological information on corticospinal tract neurons if their amplitude is decreased due to the loss of spinal motor neurons. However, we also agree with the reviewer that TST may represent a useful solution in this context; this information has been added to the manuscript (lines 482-484).

The Mmax is also influenced by skin preparations and electrode placement.

We certainly agree with this notion, but since these are technical factors that influence all neurophysiological measures, we do not think that mentioning them would enrich our discussion.

'Impaired' SICI. Maybe 'reduction in SICI' ?

The text has been modified as suggested.

In AD, reduced SAI is a relevant MEP marker indicating cholinergic deficit.

We agree with the reviewer. This has been mentioned in the text (lines 508-510)

Narrowing down the MEP interpretations by combining them with complementary measurements

Here twitch force recordings should be mentioned as they are not subjected to phase cancellations.

We thank the reviewer for pointing this out. This information has been added to the text (lines 555– 560).

The authors fail to make a convincing case how TEPs can narrow down MEP interpretations. This reviewer would argue that TEPs rather broaden the interpretations as the two readouts are very poorly related. It is unclear from the paragraph, how the TEP read-out contributes to our understanding of the MEP.

By using the wording “narrowing down the MEP interpretations”, we were referring to the fact that MEP can be influenced by spinal excitability. By contrast, measures not dependent on spinal cord function, such as TEPs, allow for collecting more specific information on M1 function. We agree with the reviewer, however, that this probably does not constitute a narrowing down of MEP interpretation. To clarify this, we changed the title of the paragraph; the new one is “Isolating information on M1 physiology by means of additional measures”.

Rather than referring to general review papers, the authors need to focus on TEP studies that directly relate MEP read-outs with TEP read-outs that have been (source) localized to the precentral gyrus.

We agree on this. Several studies and related information have been added to the text, following the reviewer’s comments in the following points. We thank the reviewer for his/her useful suggestions in this regard (610-624).

The relationship between precentral TEP and MEP is a matter of debate, in part due to the contamination from peripheral activation. The authors are referred to the recent paper by Biabani et al (2021) in J Physiol.

We agree with the reviewer. The mentioned study by Biabani and coworkers has been added to the references about possible contamination of TEPs by afferent activity (line 594).

Here the authors need to discuss the drawbacks of the TEP. It is heavily influenced by off-target effects that are difficult to mask and impossible to avoid. (see e.g. Conde et al. 2018, Neuroimage).

We agree with the reviewer; the section on confounding factors related to TEP interpretation has been expanded, including the mentioned reference (lines 592-603).

A way to relate EEG to MEPs is to use EEG to inform the timing of precentral TMS based on the expression of precentral oscillatory brain activity. Here, the literature is not as clear as the authors try to indicate (see Madsen et al. 2019, Brain Stim, Karabanov et al. 2021, Brain Stim). Based on a balanced view on the existing literature, one may conclude that the power or phase of pericentral cortical oscillations may account for a subtle fraction of trial-to-trial variability of the MEP amplitude, but this depends critically on methodological choices in terms of spatial filtering, inter-stimulus interval and number of stimuli.

We agree and thank the reviewer for pointing to these articles, which provide very interesting data. They have been included in the text (lines 647-655).

Another option that should be mentioned is to apply post-hoc trial sorting of MEP trials according to ERP readouts (see Verleger et al. 2006, Exp Brain Res. and Eur J Neurosci. 2009).

We agree with the reviewer that this approach represents a useful method to increase physiological information provided by MEPs. The mentioned articles have been added to the text (lines 634-637).

Referee #2:

Summary

In the current manuscript, Spampinato and colleagues review the use of motor evoked potentials (MEPs) to study excitability of the corticospinal system. The authors emphasise that the MEP is a complex measure and that interpreting changes in MEP amplitude requires careful interpretation. The topic of the review is welcome considering how widely used the MEP is, and how often narrow interpretations of changes in MEPs are adopted. In general, the review is well written, however I do have some suggestions which I think will help to further refine some of the arguments raised by the author. In particular, I was expecting a stronger focus on the MEP itself - how is it measured, what does it look like, and how to changes in output from the motor cortex, spinal excitability, temporal dispersion etc impact MEP size and shape? I've detailed some of my suggestions below. I leave it up to the authors whether they choose to adopt them, however I do think they will further improve an already excellent review.

Major Comments

1. Given the title, I was expecting to read a more detailed account of the MEP itself, what is typically measured, and what can impact those measurements. Instead, there is a lot of discussion on basic physiology of TMS and motor output in terms of D-waves and I-waves, etc. These sections are really nice, and well written, but I found it a little unclear as to how changes to these outputs would impact the MEP. Furthermore, while most of the review seems to focus on the amplitude of the MEP, less space is given to other measures such as the threshold for evoking an MEP and the latency of the MEP. This is just a suggestion, but the authors might like to consider restructuring the first section to detail what an MEP is, how it is measured and what it typically looks like (possibly with an accompanying figure focusing specifically on the MEP and what is typically measured). They could then detail what is typically measured from an MEP (motor threshold, latency, peak-to-peak amplitude, area under the curve) and what is known to impact each of these measures. For example, the presence or absence of D-waves can alter the MEP latency, which changes as stimulation intensity is increased; or differences in scalp-to-cortex distance explains a lot of the variance in motor threshold between individuals however this measure is also sensitive to changes membrane potentials governed by sodium-gated ion channels as shown in pharmacological work (just as two examples). By detailing what can impact the properties of the MEP earlier in the manuscript, this will help to contextualise the two practical examples explained in the second half of the review (e.g., changes with motor control and in neurological disorders).

We thank the reviewer for suggesting the inclusion of other commonly measured characteristics of the MEP, such as the latency, motor threshold, and area under the curve. We have primarily focused on the MEP amplitude due to its popularity in many basic research and behavioral-TMS designs but agree that it is important to highlight the importance of other MEP characteristics, as they may be more useful than the amplitude in the context of clinical settings. As such, we have now added a new figure (Fig 2) that describes these different components, and (as suggested by the reviewer) we have also added what can impact these measures both in the figure caption and throughout the main text. We believe that this addition now provides more contextual information that will help guide the reader for the 2nd half of the review.

2. I'd suggest expanding the description and importance of desynchronised excitation of motor units in determining the size and shape of the MEP. I'd pay particular attention to the article by Magistris et al 1998, *Brain* as an example of how phase cancellation can impact MEP amplitudes. This is a crucial concept for understanding the size and shape of the MEP, especially when considering pathological impacts on the MEP (e.g., from demyelination). It might be helpful to compare the MEP size and shape to that of the compound muscle action potential (a good example is in Groppa et al 2012, *Clin Neurophysiology*). This is alluded to later in the review, but could be introduced earlier. Furthermore, it is important to fully explain what a compound muscle action potential is, and how it is generated/measured.

We thank the reviewer for their concern and agree that it is important to describe the details of desynchronized excitation of motor units and how this can impact the MEP size/amplitude (see new subsection "Measuring MEPs: Dispersion"). In addition, we define what CMAPs are and how they can be measured in humans, as well as their importance for being compared to MEPs in the triple pulse stimulation technique.

3. Pharmacological studies have taught us a lot about the different cortical mechanisms that MEPs are sensitive to, however this body of literature is barely discussed. I suggest including a more thorough description of how MEPs are altered by different pharmacological agents. Given that MEP amplitudes are altered by GABAergic drugs, this might help better contextualise the section discussing how MEPs are altered during movement preparation.

We agree with the reviewer that it is important to highlight previous in the TMS literature that has incorporated pharmacological interventions. We have now added information regarding the pharmacological effects on I wave (see lines 147-148), on distinct elements of the MEP (Figure 2 caption), and how they influence the response of paired-pulse TMS measures (see lines 313-314).

4. I was a little puzzled by the paragraph on MEP variability. Statements like 'One of the main challenges is dealing with relatively large trial-by-trial MEP variability' suggest that MEP variability simply reflects noise. I think the complexities of MEP variability are under-appreciated within the field and perhaps warrant a slightly more nuanced discussion, especially considering the title of the review. For example, the approach of taking the average MEP as a summary statistic ignores the fact that variability in MEP amplitude within an individual is rarely normally distributed, which raises the question as to whether this widely used approach is appropriate. Furthermore, the characteristics of MEP variability change with different stimulation intensities (from right skewed to left skewed and can be bimodal at some intensities; Goetz et al 2014 *Brain Stimulation*), with different states (e.g., at rest vs during a muscle contraction) and with different pathological disorders (example in MS - Britton et al 1991). Also, MEP amplitudes change over time and with different interstimulus intervals (e.g., Julkunen et al 2012 *Brain Stimulation*). All of these characteristics raise a lot of questions about how MEPs are measured and whether we miss a lot of critical information by simply taking the mean of the MEP at one stimulus intensity as opposed to more accurately characterising MEP variability. I think expanding the discussion a little will help to highlight the challenges of how best to measure the MEP given that it is variable, and challenge the field to think a bit more about what can we learn from this variability.

We thank the reviewer for their very helpful and important suggestion. The entire section regarding MEP variability has now been expanded to include some criticism regarding how averaging MEP amplitudes from a small sample size of pulses (10-15) should no longer be considered a valid standard, as it does not accurately nor reliably characterize cortical excitability (lines 204-207). We also highlight how variability should not necessarily be considered "bad" but rather informative for explaining cross-individual differences and we also have included 'Table 1' which covers factors that influence the MEP, along with current solutions to limit their effects on the MEP.

5. The paragraph on how changing coil angle impacts motor output would benefit from some more detail. PA vs AP changes the I-wave composition - how does this impact MEPs? AP is thought to stimulate premotor terminals - what about PA stimulation? What is the current consensus on where this stimulates?

We thank the reviewer for bringing this to our attention. We feel that the re-structuring of the document now makes it easier for the reader to understand how applying different current directions over M1 impacts the MEP (discussion regarding latency and stimulus intensity). Additionally, we now discuss how it remains debated what M1 TMS exactly stimulates. The latest modeling work suggests that sections of the premotor cortex (PMd) may be stimulated with both PA and AP-currents; however, AP currents appear to target neurons that are slightly more anterior to those stimulated with PA currents (see lines 250-261).

6. In the TMS-EEG section, I suggest also discussing some of the limitations of this method (e.g., the recent debates regarding how much of the TEP represents sensory input with certain experimental arrangements - Gordon et al 2018, Conde et al 2019, Biabani et al 2019, Rocchi et al 2020 etc.).

We thank the reviewer for this comment. The section on limitations of TMS-EEG has been expanded accordingly (lines 587-608).

Minor comments

1. Introduction: 'Finally, we discuss how the combination of TMS with electroencephalography (EEG) may represent as a possible strategy to more thoroughly study cortical components of the MEP.' I would suggest rewording this sentence. If I understand, I think what the authors are trying to convey here is that TEPs may represent an alternative measure for indexing cortical excitability similar to the MEP, but without the confound of spinal excitability. TEPs aren't really the cortical components of MEPs - they are sensitive to different spatial scales than MEPs (which focus on the output of a very small minority of corticospinal neurons, which is pointed out later in the review). But both may be sensitive to changes in cortical excitability.

We agree with the reviewer. It is indeed more correct to state that TEPs may be useful to investigate motor cortical excitability without the confounding factor represented by the activation of spinal motor circuitry, with the limitations mentioned later in the text. The sentence has been reworded accordingly (lines 70-72).

2. Basic principles of TMS: 'minimal or no discomfort' - this is highly dependent on where the stimulation is applied. In some regions there can be some discomfort due to the stimulation of face/scalp muscles or nerves. I'd suggest rewording to reflect this.

This phrase has now been removed.

3. 'Theoretically, this implies that tweaking the pulse parameters is one way to reduce the variability of MEP responses across individuals; however, it is important to stress that the ongoing activity of M1 neuronal populations will ultimately dictate the MEP response.' Is there any data to back this up? If not, I'd suggest removing as is overly speculative.

We agree that this sentence is not appropriate and have now removed it from the text.

4. 'Finally, it is important to note that since the MEP represents the output of a small population of the entire pyramidal tract output, it is possible to make statements about the focality of effects in a way that has more resolution than fMRI or EEG. If one is able to conclude that the changes in MEP are due to changes

in M1, the immediate secondary conclusion is that the changes in M1 are caused by changes in its input, therefore allowing one to infer changes happening in other, connected, parts of the motor circuits.' I found this paragraph confusing and I'd suggest rephrasing. I'm a little unclear what is meant by 'focality' and 'resolution' in this context. Are the authors suggesting that, due to the input-output nature of MEPs and the specificity to one muscle that a more causal inference is possible? How are changes located to M1 (presumably by ruling out spinal mechanisms)? Note the reliance on motor output is both a strength and weakness - it does allow a higher degree of specificity, but is highly limited to motor systems.

After restructuring the original manuscript to fit all of the suggestions by the reviewers, we have decided to remove this paragraph from the text as it was likely more confusing than helpful for the reader to move on towards the second half of the review.

5. Figure 2B: The cartoon depicting surround inhibition could be a little more accurate. It's not clear that activation of this neuron suppresses other connected neurons (or vice versa - are the closed or open triangles representing inhibition)?

We apologize for this confusion and have now edited this figure to make it clearer that a particular neuron may be activated by certain interneurons, while other PTN may be inhibited to prevent the activation of task-irrelevant or antagonist muscle (note: Figure 2 is now Figure 4).

Dear Dr Spampinato,

Re: JP-TR-2022-281885X "Motor potentials evoked by transcranial magnetic stimulation: interpreting a simple measure of a complex system" by Danny Adrian Spampinato, Jaime Ibanez Pereda, Lorenzo Rocchi, and John C Rothwell

Thank you for submitting your Topical Review to The Journal of Physiology. It has been assessed by a Reviewing Editor and by 2 expert referees and I am pleased to tell you that it is considered to be acceptable for publication following satisfactory revision.

The reports are copied at the end of this email. Please address all of the points and incorporate all requested revisions, or explain in your Response to Referees why a change has not been made.

NEW POLICY: In order to improve the transparency of its peer review process The Journal of Physiology publishes online as supporting information the peer review history of all articles accepted for publication. Readers will have access to decision letters, including all Editors' comments and referee reports, for each version of the manuscript and any author responses to peer review comments. Referees can decide whether or not they wish to be named on the peer review history document.

I hope you will find the comments helpful and have no difficulty in revising your manuscript within 4 weeks.

Your revised manuscript should be submitted online using the links in Author Tasks Link Not Available. This link is to the Corresponding Author's own account, if this will cause any problems when submitting the revised version please contact us.

You should upload:

- A Word file of the complete text (including any Tables);
- An Abstract Figure, (with accompanying Legend in the article file)
- Each figure as a separate, high quality, file;
- A full Response to Referees;
- A copy of the manuscript with the changes highlighted.
- Author profile. A short biography (no more than 100 words for one author or 150 words in total for two authors) and a portrait photograph of the two leading authors on the paper. These should be uploaded, clearly labelled, with the manuscript submission. Any standard image format for the photograph is acceptable, but the resolution should be at least 300 dpi and preferably more.

- A 'Cover Art' file for consideration as the Issue's cover image;
- Appropriate Supporting Information (Video, audio or data set https://jp.msubmit.net/cgi-bin/main.plex?form_type=display_requirements#supp).

To create your 'Response to Referees' copy all the reports, including any comments from the Senior and Reviewing Editors into a Word, or similar, file and respond to each point in colour or CAPITALS. Upload this when you submit your revision.

I look forward to receiving your revised submission.

Yours sincerely,

Professor Laura Bennet
Senior Editor
The Journal of Physiology
<https://jp.msubmit.net>
<http://jp.physoc.org>
The Physiological Society
Hodgkin Huxley House
30 Farringdon Lane
London, EC1R 3AW
UK
<http://www.physoc.org>
<http://journals.physoc.org>

EDITOR COMMENTS

Reviewing Editor:

As a Special Case Resubmission, the manuscript has been assessed by two referees who did not view the original version of the article. Both of the reviewers see considerable merit in the manuscript, although some additional steps may be necessary to highlight those aspects that provide an advance upon recent reviews of a similar nature. You will also see that both referees have several recommendations concerning ways in which the presentation could be made more precise, comprehensive, and thus of greater utility for the naive reader. There are also several instances in which additional (sometimes seminal) citations should be included.

In summary, while the consensus is that the review has the potential to be influential, the impact of the paper is likely to be enhanced by accentuating those aspects of the analysis that constitute truly novel contributions. In this vein, it may be advantageous to consider whether there are any other unique insights that might have emerged in the course of writing and revising the review. In a somewhat similar vein, the authors might consider whether there remains scope for greater of analysis. The comments of the referees provide several suggestions in this regard.

Senior Editor:

Thank you for your submission. The reviewers have provided in depth comments that should be addressed to help highlight aspects of the review which are truly novel as per the remit of the topical reviews for The Journal (versus other reviews and chapters on the subject already published). To help facilitate this, reviewers have suggested a deeper dive in places in terms of analysis of the subject matter.

REFeree COMMENTS

Referee #3:

The manuscript discusses recent discoveries about motor potentials in TMS. The text provides a very good overview and should be published. However, a number of serious gaps need to be closed to ensure high impact.

- Please add a list of acronyms. The text is very hard to read even for someone from the field. However, reviews are often read a lot also by young researchers. Please generally improve readability with respect to acronyms. Many acronyms can be spelled out as there is no point in making it shorter in print at the cost of readability (in contrast to maybe hand-written texts).
- It would be very interesting to also include invasive recordings or work (in the cortex, spine, any maybe just single-unit recording in muscles) to link it to the indirect observation of surface EMG.
- I missed a bit a discussion of the different role of cortical versus spinal mechanisms and influences. I would even suggest adding a figure that shows all influences from the cortical to the spinal level and potential dynamics at which each might change.
- The summary and the table suggest the pulse duration as an important factor if not a solution to control variability. However, the text does not cover this topic at all. That gap should be filled.
- Also the pulse itself (such as monophasic, biphasic and so on), which apparently has large influence on the outcome and has been studied intensively, is practically ignored.
- The text claims that TMS pulses are approximately 1 ms long. Most TMS pulses are rather shorter than that. The field is then given as 100-200 μ s long, which is shorter than most pulses of available machines. The field is just as long as the pulse.
- On pg. 4, the text suggests that at least two synapses have to be activated for an MEP and mentions one in the cortex. I assumed that the shortest possible latency bypasses any cortical synapse and activates the pyramidal axon directly.
- Later, in the last paragraph on that page, it is said that various potentials might cancel each other. I would say that they rather smoothen and wash out each other to a longer sum potential.
- Please consistently add spaces between numbers and their units (e.g., 5 ms).
- In many places, the text claims something but does not provide any source. For example, 90% of the total fibers are called small diameter fibers. Where is that documented?
- The last sentence of that paragraph needs punctuation (after Wiegel et al.).

- The text discusses qualities of MEPs and recruitment. However, it does not mention the rather steep exponential growth with increasing stimulation strength documented in various IO curve papers. Increasing the pulse by x% does not let the MEP grow proportionally. Thus, stimulating a bit stronger does apparently not just add a few more fibers. This steep growth may appear to control some gain even. That might be worthwhile a longer discussion in the paper, which I missed.
- The manuscript shortly mentions the high variability and the odd statistical distribution of MEPs. This topic might benefit from more details as it might shed light on mechanisms. Furthermore, it has quite an impact on technicalities just mentioned (without highlighting the link) a bit later. MEPs do not average well due to the highly skewed distribution. The average is not well defined and dominated by a few extremely large MEPs that show up very rarely.
- Please provide references for statements such as "modelling work suggests that PA TMS activates a boundary region between caudal PMd and anterior M1" or corticospinal "neurons that originate from this region rarely have monosynaptic connections to spinal motoneurons" or "AP currents are thought to activate a spatially segregated premotor neural population". In general, there are quite many statements that would benefit from a specific source. Those might be given elsewhere already, but a reader would have a hard time going through the entire literature list trying to find maybe one paper that backs the specific statement.
- It might be interesting to also mention and discuss the influence of contraction (pre, post) on TMS-induced excitability changes. This effect is well documented but explanations seem to be missing. Could that be translated to other targets?
- The first half of page 10 has no references but many statements that would deserve a source.
- This page also mentions states of the motor cortex and the influence of active use. How would motor imagery and mirror activation without muscle contraction compare to it? What is actually happening in the motor cortex in those cases compared to a clear contraction based on a corticospinal signal? Are different neurons activated or is the signal blocked from traveling on at some point and where?
- Do refractory effects play a role here?
- Page 11 may also deserve a few more sources for statements, for example on MEP changes based on ISI.
- It is mentioned that peripheral nerve stimulation (as then likely used in PAS, although PAS is not covered) might influence MEPS. Since such techniques are not widely done, they would deserve more specifics about the procedure and what is done there.
- The text mentions multi-locus TMS as a nice technique to perform stimulation at two sites without repositioning. However, there is also literature performing such experiments with multiple coils, even very closely together, which may be worthwhile discussing in that context.
- The text refers to changes of the membrane resistance in line 408. Does that mean the axon? Could you refer to literature for that and elaborate a bit in the text?
- Whereas the variability of MEPs was at least mentioned before, the text does not cover the reproducibility and variability of TEPs (both intra- and interindividual).
- TEPs as usually recorded have the technical problem that they use extremely short ISIs for sufficient repetitions to average out noise and variability. Such timings are known to lead to large correlation between responses in MEPs. Such technicalities would be worth mentioning.
- Furthermore, the interpretation of TEP components (and also what we do not understand) would benefit from more clear statements. To my understanding, there is still lots of debates going on for anything beyond the attention, somatosensory and auditory responses.
- Please provide references in line 625 (TEP signal from neurons perpendicular to the scalp). To my understanding that is not more than an assumption without any experimental backing (which would be hard) as also the generation of EEG signals is to a large part not as clearly assigned to specific neural elements as sometimes claimed in the class room.
- What can be said about the relationship between (brain) state and EEG? Would the authors dare to define the brain state for the motor system?
- The summary could ideally really summarize some key discoveries and insights about MEPs
- Fig. 1 misses a closing bracket.
- Fig. 1 seems to explain early and late I waves mostly through different distances of synapses to the soma of the pyramidal neuron. How would that explain the rather clear timings between the individual waves? Furthermore, showing where ML coil orientation primarily activates would be interesting in this figure.

- Line 727 mentions a metric. Which metric is meant here?

- In the table, I would add the ISI as an external factor, which has large influence due to the relatively long-lasting correlation between pulses. The duration of such interaction effects may also indicate how long a signal injected with TMS may circulate in cortical (and maybe spinal) circuits.

Referee #4:

This is a review on motor evoked potentials (MEP) from transcranial magnetic stimulation (TMS) and also covers the topic of TMS-EEG. I do not have major criticisms for this review. However, I am not clear how much this adds to other recent reviews, book chapters and books on similar topics.

I have some specific comments.

Line 76 - It was stated that the magnetic field produced by TMS is " ~ 1 ms" in duration. The duration is actually much shorter, closer to ~ 100 μ s in duration.

Line 100 - "at least two CNS synapses..". It is not always correct. For example, lateral-medial current can activate pyramidal neurons directly leading to the D-wave.

Lines 208-212 - For the number of trials needed, other studies (e.g. PMID: 28264713) have produced different results. One also has to consider practical aspects of studies such as subject (and experimenter) fatigue in determining the number of trials for a study.

Line 245 - "to facilitate the monosynaptic H-reflex.." A reference should be provided for this statement.

Line 294 - SICI is more commonly referred to as short-interval intracortical inhibition (e.g. guideline articles from the International Federation of Clinical Neurophysiology)

Lines 297-298 - "GABA agonists". I believe the authors are referring to benzodiazepines. They are not GABA agonists because they do not bind to the GABA receptor site, but are positive allosteric modulators of the GABA-A receptor.

Line 325 - Does opposite effects of benzodiazepine on SICI vs SAI and LAI indicates that they are mediated by GABAergic circuits? Could it be opposite effect on the same circuit?

Lines 404-405 - "MEP amplitude only increase a few tens of milliseconds.." Other studies have found different results with MEP amplitude increase occurring earlier (e.g. PMID: 16932969, PMID: 34234660, PMID: 9749597). While different experimental design and analyzes may account for some of the differences, these results should be mentioned.

Lines 462-462 - It is unclear what type of future work the authors are suggesting.

Lines 468-469 - What are the "advanced neuromodulation techniques"?

Line 541 - Previous studies have shown steeper input-output relationship in Parkinson's disease (PD) at rest but less steep during muscle activation (PMID: 8164834).

Lines 542-555 - Multiple studies have reported the abnormalities described by the authors. While it is not practical to cite all the studies, I believe that the first study that showed the abnormality should be referred to. For example, in PD decreased SICI was first described by Ridding et al (PMID: 7847860), increase in SICI with deep brain stimulation was first described by Cunic et al (PMID: 12058096) and increased SICF was first described by Ni et al. (PMID: 23576626).

Line 545 - A large study has shown that SICI is abnormal in the less affected hemisphere in PD (Ammann et al, PMID: 33141146).

Line 653 - "reduce individual responsiveness to TMS" TMS responses when delivered at some EEG phases are increased rather than reduced (e.g. PMID: 31631058)

Line 673 - "how activity spreads from M1 to other areas of the cortex". One of the advantages of TMS-EEG compared to

MEP measurement is that responses to TMS from cortical areas other than the M1 can be measured. This was not mentioned.

Line 701 - "diverse" should probably be "different"

Line 707-708. The authors used "motoneuron" in the figure but "motor neuron" in the figure legend. Both can be used but the authors should use one term consistently.

Fig 3A. The figure depicts LICI inhibiting SICI. This should be discussed and appropriate references cited.

Fig 4. I find this figure confusing. What do the different colors for the pyramidal tract neurons represent? In (A) and (C), only one of the PTNs has connections to the interneurons but in (B) all 4 of them have connections - why? How the interneurons led to inhibition of all 4 PTNs in (A) and only one PTN in (B) are not clear. I do not see "different current directions" being depicted in (C).

REQUIRED ITEMS:

-Your MS must include a complete "Additional information section" with the following 4 headings and content:

Competing Interests: A statement regarding competing interests. If there are no competing interests, a statement to this effect must be included. All authors should disclose any conflict of interest in accordance with journal policy.

Author contributions: Each author should take responsibility for a particular section of the study and have contributed to writing the paper. Acquisition of funding, administrative support or the collection of data alone does not justify authorship; these contributions to the study should be listed in the Acknowledgements. Additional information such as 'X and Y have contributed equally to this work' may be added as a footnote on the title page.

It must be stated that all authors approved the final version of the manuscript and that all persons designated as authors qualify for authorship, and all those who qualify for authorship are listed.

Funding: Authors must indicate all sources of funding, including grant numbers. If authors have not received funding, this must be stated.

It is the responsibility of authors funded by RCUK to adhere to their policy regarding funding sources and underlying research material. The policy requires funding information to be included within the acknowledgement section of a paper. Guidance on how to acknowledge funding information is provided by the Research Information Network. The policy also requires all research papers, if applicable, to include a statement on how any underlying research materials, such as data, samples or models, can be accessed. However, the policy does not require that the data must be made open. If there are considered to be good or compelling reasons to protect access to the data, for example commercial confidentiality or legitimate sensitivities around data derived from potentially identifiable human participants, these should be included in the statement.

Acknowledgements: Acknowledgements should be the minimum consistent with courtesy. The wording of acknowledgements of scientific assistance or advice must have been seen and approved by the persons concerned. This section should not include details of funding.

END OF COMMENTS

Dear Reviewers,

Thank you for the positive reviews and insightful feedback on our review article. We have addressed the concerns brought up by each reviewer and believe that the manuscript has dramatically improved thanks to their input. As is seen in the manuscript. We have now added several missing citations to the main text and have added some new paragraphs with the discussion suggested by the reviewers. Below, our responses to the reviewers' concerns can be seen in the bolded text.

Referee #3:

The manuscript discusses recent discoveries about motor potentials in TMS. The text provides a very good overview and should be published. However, a number of serious gaps need to be closed to ensure high impact.

- Please add a list of acronyms. The text is very hard to read even for someone from the field. However, reviews are often read a lot also by young researchers. Please generally improve readability with respect to acronyms. Many acronyms can be spelled out as there is no point in making it shorter in print at the cost of readability (in contrast to maybe hand-written texts).

We thank the reviewer for pointing out this issue of clarity. A list of acronyms has now been added to the main text.

- It would be very interesting to also include invasive recordings or work (in the cortex, spine, any maybe just single-unit recording in muscles) to link it to the indirect observation of surface EMG.

We thank the reviewer for the suggestion of adding some invasive recordings. We have now added some information from seminal studies conducted in both animal and human work that includes single-cell, spinal, and single-motor unit recordings in sections that discuss how MEPs are produced and when describing the recruitment of different I-waves with directional TMS. The text now includes: "Experimental approaches in rodents and non-human primates recording single corticospinal cell responses to TMS have shown that a single pulse of TMS evokes a cascade of high-frequency activity in the stimulated region (Mueller et al., 2014; Li et al., 2017). In humans, a suprathreshold TMS pulse evokes (i) multiple descending corticospinal volleys (as shown by invasive epidural electrodes placed over the high cervical cord) (Di Lazzaro et al., 2004) and (ii) multiple peaks of increased firing in the post-stimulus time histograms (recorded from single motor units of the targeted muscle) (Day et al., 1989)."

- I missed a bit a discussion of the different role of cortical versus spinal mechanisms and influences. I would even suggest adding a figure that shows all influences from the cortical to the spinal level and potential dynamics at which each might change.

Throughout the 1st section of the text, we have attempted to discuss how the MEP is a signal that is composed of cortical and spinal mechanisms. We believe that the cortical mechanisms have been discussed thoroughly in this text (e.g. I-waves) and would like to refer the section "*Towards a real model of MEP production: multiple pathways*" to the reviewer, which highlights how corticospinal neurons synapse with various types of neurons in the spinal cord. We agree with the reviewer that some details of spinal

mechanisms influencing the MEP could be added to the text and how this leads to the problem of dispersion and variability. We have opted not to add a figure as we are not sure that it is possible to depict all the potential cortical and spinal influences in a single figure in a clear manner.

Rather, we have made some significant changes to the text, for instance, in the “Measuring MEPs: dispersion” section, we have now added to the introduction the following sentences: “The size of the MEP largely depends upon fluctuations in spinal motoneuron excitability and the distribution of synaptic activation across the motoneuronal pool. As discussed, the descending cortical volleys are capable of influencing motoneuron activation through two pathways: (1) monosynaptic connections to motoneurons (Lemon, 2008); (2) polysynaptic connections via projections to spinal interneurons that, in turn, synapse to motoneurons (Nielsen et al., 1993; Pierrot-Desilligny, 2002). Depending on these influences, some motoneurons may not fire or may even discharge multiple times, leading to a desynchronized discharge (Groppa et al., 2012).”

In the “Measuring MEP: Variability” section, we also cover how different types of influences (e.g. mental, spinal cord and somatosensory inputs) can play an important role in influencing trial-to-trial MEP responses.

- The summary and the table suggest the pulse duration as an important factor if not a solution to control variability. However, the text does not cover this topic at all. That gap should be filled. & - Also the pulse itself (such as monophasic, biphasic and so on), which apparently has large influence on the outcome and has been studied intensively, is practically ignored.

We thank the reviewer for pointing out these important topics regarding pulse duration and pulse waveform. We agree that these are important concepts to be discussed in some detail and as such, we have now added a new paragraph under the section “Recruitment of different I-waves by changing the TMS orientation, pulse and waveform” to address this concern.

- The text claims that TMS pulses are approximately 1 ms long. Most TMS pulses are rather shorter than that. The field is then given as 100-200 μ s long, which is shorter than most pulses of available machines. The field is just as long as the pulse.

This mistake has now been modified as suggested by the reviewer.

- On pg. 4, the text suggests that at least two synapses have to be activated for an MEP and mentions one in the cortex. I assumed that the shortest possible latency bypasses any cortical synapse and activates the pyramidal axon directly.

We thank the reviewer for this important point. We have now adjusted the text to be more precise.

- Later, in the last paragraph on that page, it is said that various potentials might cancel each other. I would say that they rather smoothen and wash out each other to a longer sum potential.

We have now adjusted the text as recommended by the reviewer, and reads as “Thus, their negative and positive peaks would occur at different times and thus they may smoothen and wash out each other to a longer sum potential.”

- Please consistently add spaces between numbers and their units (e.g., 5 ms).

We have revised the manuscript to maintain consistency in the spaces between numbers and their units.

- In many places, the text claims something but does not provide any source. For example, 90% of the total fibers are called small diameter fibers. Where is that documented?

We apologize for not placing a source after certain statements, like the one the reviewers have pointed out. After reading through the text, we have now added appropriate citations to statements that were previously lacking an appropriate citation.

- The last sentence of that paragraph needs punctuation (after Wiegel et al.).

Fixed.

- The text discusses qualities of MEPs and recruitment. However, it does not mention the rather steep exponential growth with increasing stimulation strength documented in various IO curve papers. Increasing the pulse by x% does not let the MEP grow proportionally. Thus, stimulating a bit stronger does apparently not just add a few more fibers. This steep growth may appear to control some gain even. That might be worthwhile a longer discussion in the paper, which I missed.

We thank the reviewer for bringing this lack of clarity in the text to our attention. In the second paragraph of the second section "Measuring MEP: variability" we have now added more details about stimulus-response curves and relevant sources for the readers. The text now reads: "The input-output curves reflect the gain in MEP amplitude with increasing stimulus intensity, which grows exponentially and eventually plateaus to saturation at high-intensity levels. In other words, increasing the intensity does not simply increase the number of recruited corticospinal fibers. It also influences the temporal dispersion of spikes propagating along the corticomotor pathways (Rossini et al., 2015)."

- The manuscript shortly mentions the high variability and the odd statistical distribution of MEPs. This topic might benefit from more details as it might shed light on mechanisms. Furthermore, it has quite an impact on technicalities just mentioned (without highlighting the link) a bit later. MEPs do not average well due to the highly skewed distribution. The average is not well defined and dominated by a few extremely large MEPs that show up very rarely.

As highlighted by the reviewer, the topic of MEP variability is important to discuss in the frame of this review. We have now added more details regarding this. We have significantly modified the paragraph discussion of the distribution of MEP amplitudes and have added potential neurophysiological processes that may influence these responses.

- Please provide references for statements such as "modelling work suggests that PA TMS activates a boundary region between caudal PMd and anterior M1" or corticospinal "neurons that originate from this region rarely have monosynaptic connections to spinal motoneurons" or "AP currents are thought to activate a spatially segregated premotor neural population". In general, there are quite many statements that would benefit from a specific source. Those might be given

elsewhere already, but a reader would have a hard time going through the entire literature list trying to find maybe one paper that backs the specific statement.

We apologize for not having references to these states, we have now fixed this in the new version.

-It might be interesting to also mention and discuss the influence of contraction (pre, post) on TMS-induced excitability changes. This effect is well documented but explanations seem to be missing. Could that be translated to other targets?

We thank the reviewer for their suggestion regarding the influence of muscle contraction. Due to the length of this manuscript, we have decided to limit our discussion to recording and comparing MEPs at rest vs. muscle contraction in the section on MEP variability. We have provided a citation of a previous important article that readers can refer to if they would like more information regarding the influence of contraction.

- The first half of page 10 has no references but many statements that would deserve a source.

Fixed.

- This page also mentions states of the motor cortex and the influence of active use. How would motor imagery and mirror activation without muscle contraction compare to it? What is actually happening in the motor cortex in those cases compared to a clear contraction based on a corticospinal signal? Are different neurons activated or is the signal blocked from traveling on at some point and where? Do refractory effects play a role here?

The activity of the central versus system when undergoing motor imagery/mirror activation when compared to muscle contraction should reveal some important differences. While one might not expect activity in the corticospinal tract during motor imagery, recent work has shown imagery does in fact produce some voluntary drive along the corticospinal tract (doi: 10.1152/jn.00952.2015); however, this activity does not recruit or activate alpha-motoneurons, which would be activated in a case of muscle contraction. This study also demonstrated that motor imagery does activate low-threshold spinal structures (presynaptic interneurons), so it appears that different neurons (even within the spinal network) are activated in these scenarios. As motor imagery is a topic that goes beyond the scope of this review, we have elected not to include this information in the manuscript; however, we have now mentioned that state-dependent effects are also influenced by “mental training of a motor behavior”.

- Page 11 may also deserve a few more sources for statements, for example on MEP changes based on ISI.

We thank the reviewer for pointing this out to us, and we have added more references in this section.

- It is mentioned that peripheral nerve stimulation (as then likely used in PAS, although PAS is not covered) might influence MEPS. Since such techniques are not widely done, they would deserve more specifics about the procedure and what is done there.

We agree that many readers of this manuscript may not be familiar with peripheral nerve stimulation and thus have added more details about the procedure. The text now reads as, “Electrical or mechanical stimulation of a particular peripheral nerve produces afferent activity (sensory input) from the contralateral limb that reaches M1 through thalamo-cortical afferents or the somatosensory cortex (Hamada et al.,2012). The effect of sensory stimulation on the MEP amplitude depends on the time between electrical stimulation of the targeted nerve and a supra-threshold TMS pulse over M1 (Tokimura et al., 2000). MEP suppression is observed when the ISI between nerve stimulation and M1 TMS is either short (20-25 ms, short-latency afferent inhibition: SAI) or long (200-1000 ms, long-latency afferent inhibition: LAI). Importantly, these effects occur only if the homotopic stimulation of sensory input matches the location of the muscle targeted by TMS.”

- The text mentions multi-locus TMS as a nice technique to perform stimulation at two sites without repositioning. However, there is also literature performing such experiments with multiple coils, even very closely together, which may be worthwhile discussing in that context.

We thank the reviewer for their suggestion for adding literature that has successfully used twin-coil TMS approaches in brain regions that are quite close together. The text now reads, “The twin coil TMS approach has allowed the field to investigate the physiological interactions of M1 with other brain regions, such as bilateral posterior parietal cortices, ventral (PMv) and dorsal (PMd) premotor cortices, supplementary motor area, and the cerebellum. Several studies have shown that small-diameter coils are generally needed for these interactions, along with careful coil positioning for the cortical sites that are very close together (Civardi et al., 2001; Koch et al., 2008; Davare et al., 2009).”

- The text refers to changes of the membrane resistance in line 408. Does that mean the axon? Could you refer to literature for that and elaborate a bit in the text?

We thank the reviewer for bringing this point of clarity to our attention. We have now added some detail regarding changes in membrane resistance in the context of motor processing, the text now reads, “At the cellular level, for example, increases in synaptic activation in a given cell during certain motor tasks may inherently lead to changes (decays) in its membrane resistance (Paulus and Rothwell, 2016). This would imply that additional transmembrane currents generated by synaptic inputs would have a smaller effect on the neural discharge rate. In other words, an increase in membrane conduction during the intense activation of a neuron may reduce MEP responses stronger than one expects.”

- Whereas the variability of MEPs was at least mentioned before, the text does not cover the reproducibility and variability of TEPs (both intra- and interindividual).

We thank the reviewer for this important comment. Under the second paragraph of theTMS-EEG section, we have now added the following sentence: “Another advantage is that TEPs have high interindividual reproducibility when stimulation is given over both motor and premotor cortices (Lioumis et al., 2009; Kerwin et al., 2018) and low levels of individual variation across multiple sessions (Matamala et al., 2018; ter Braack et al., 2019).”

- TEPs as usually recorded have the technical problem that they use extremely short ISIs for sufficient repetitions to average out noise and variability. Such timings are known to lead to large

correlation between responses in MEPs. Such technicalities would be worth mentioning.

We thank the reviewer for their concern regarding the short timing between pulses when recording TEPs. Recent work has addressed this issue by testing whether several (100) pulses with a short inter-pulse trial (1.1.- 1.4 ms) produce changes in TEPs and global mean field amplitude (doi.org/10.1515/tnsci-2022-0235). They found that these responses did not change throughout their analysis, when comparing different chunks of trials (e.g. the first 30 trials to the last 30 trials), which suggests that short trials can be used to significantly reduce the duration of TMS-EEG studies without the risk of inducing potential changes related to the short stimulation rate. We have added this to the discussion of 3rd paragraph of the TMS-EEG section.

- Furthermore, the interpretation of TEP components (and also what we do not understand) would benefit from more clear statements. To my understanding, there is still lots of debates going on for anything beyond the attention, somatosensory and auditory responses.

We apologize that we are a bit confused by the reviewer's wording in this comment and hope the following answer suffices the reviewer's point. Somatosensory and auditory evoked responses are the topics where most of the debate lies at the moment in the TMS-EEG field. Because of these artifacts, there is quite a bit of debate in disentangling true cortical responses to TMS from those due to concomitant sensory responses. Beyond this, one major issue is that there is no general agreement on the preprocessing pipeline used (in particular) for the manner in which early TMS-locked artifacts (i.e., cranial muscle activation and voltage decay) are removed (e.g., with ICA, SSP-SIR, and so on). Some authors have stated that this may make a difference ([10.1016/j.neuroimage.2021.118272](https://doi.org/10.1016/j.neuroimage.2021.118272)) and others not ([10.3390/brainsci11020145](https://doi.org/10.3390/brainsci11020145)). We have added this information to the manuscript.

- Please provide references in line 625 (TEP signal from neurons perpendicular to the scalp). To my understanding that is not more than an assumption without any experimental backing (which would be hard) as also the generation of EEG signals is to a large part not as clearly assigned to specific neural elements as sometimes claimed in the class room.

We thank the reviewer for bringing this point to our attention, and after consideration, we have now removed this sentence. Rather, TEPs probably reflect signals from populations of neurons both oriented radially and tangentially to the scalp, depending on the component. TEP P30 seems part of a radial dipole, so it might be generated by radially oriented neurons ([10.1002/mds.28914](https://doi.org/10.1002/mds.28914)) whereas, the N45/P60 components are usually seen as a tangential dipole, so they might be generated by tangentially-oriented neurons ([10.3390/brainsci11030326](https://doi.org/10.3390/brainsci11030326))

- What can be said about the relationship between (brain) state and EEG? Would the authors dare to define the brain state for the motor system?

While understanding the relationship between brain state and EEG is an intriguing point of discussion and debate, we believe that this argument is beyond the aim of our review, as there is probably no well-defined answer for “brain state of the motor system”.

- The summary could ideally really summarize some key discoveries and insights about MEPs

As this review describes several topics in detail, including the many elements that make up MEPs, how MEP changes can be difficult to interpret in behavioral and clinical contexts, and how TMS-EEG can be used as an additional tool to understand cortical excitability, we would like to maintain our summary paragraph as we believe it captures the essence of the review (which is not necessarily about new discoveries regarding the MEP, but more so in highlighting the complexities surrounding how the brain responds to brain stimulation).

- Fig. 1 misses a closing bracket and Fig. 1 seems to explain early and late I waves mostly through different distances of synapses to the soma of the pyramidal neuron. How would that explain the rather clear timings between the individual waves? Furthermore, showing where ML coil orientation primarily activates would be interesting in this figure.

We thank the reviewer for the comments regarding Figure 1 and we have now made some slight modifications to this figure (e.g. including ML coil orientation). Moreover, we agree that I-wave physiology is much more complicated than our image depicts (for example, the inclusion of GABAergic inhibitory interneurons that influence I-wave generation is missing); however, the point of this illustration is to show that M1 TMS results in transynaptic activation of pyramidal cells and that changing TMS current directions activates different populations of interneurons.

- Line 727 mentions a metric. Which metric is meant here?

We apologize for not including “Peak-to-peak amplitude” and how now added this to the Figure 2 Caption.

- In the table, I would add the ISI as an external factor, which has large influence due to the relatively long-lasting correlation between pulses. The duration of such interaction effects may also indicate how long a signal injected with TMS may circulate in cortical (and maybe spinal) circuits.

We believe the reviewer means adjusting the inter-TMS trial intervals (and not ISIs that are used for paired-pulse protocols). We have now added “Stimulation Parameters” as a factor of variability that includes inter-trial intervals.

Referee #4:

This is a review on motor evoked potentials (MEP) from transcranial magnetic stimulation (TMS) and also covers the topic of TMS-EEG. I do not have major criticisms for this review. However, I am not clear how much this adds to other recent reviews, book chapters and books on similar topics.

I have some specific comments.

Line 76 - It was stated that the magnetic field produced by TMS is " ~ 1 ms" in duration. The duration is actually much shorter, closer to ~ 100 μ s in duration.

Fixed.

Line 100 - "at least two CNS synapses..". It is not always correct. For example, lateral-medial current can activate pyramidal neurons directly leading to the D-wave.

We thank the reviewer for this point of clarity. The text now reads "When a TMS pulse is administered with the commonly used posterior-anterior (PA) current direction, at least two synapses..."

Lines 208-212 - For the number of trials needed, other studies (e.g. PMID: 28264713) have produced different results. One also has to consider practical aspects of studies such as subject (and experimenter) fatigue in determining the number of trials for a study.

We thank the reviewer for adding this important suggestion of practical issues that can influence how many TMS pulses should be considered for a study. We have now adjusted the text that reads: "practical aspects, such as participant fatigue and discomfort of prolonged stimulation, should also be considered when selecting the number of trials for a study (Cavaleri et al., 2017)."

Line 245 - "to facilitate the monosynaptic H-reflex.." A reference should be provided for this statement.

The appropriate reference is now added to the main text.

Line 294 - SICI is more commonly referred to as short-interval intracortical inhibition (e.g. guideline articles from the International Federation of Clinical Neurophysiology)

We thank the reviewer for catching this typo. The text is now fixed as referenced above.

Lines 297-298 - "GABA agonists". I believe the authors are referring to benzodiazepines. They are not GABA agonists because they do not bind to the GABA receptor site, but are positive allosteric modulators of the GABA-A receptor.

We thank the reviewer for pointing out this mistake. The text is now changed to "SICI is thought to depend on the activity of GABA-A receptors since the administration of benzodiazepine, a positive allosteric modulator of GABA-A receptor increases SICI response"

Line 325 - Does opposite effects of benzodiazepine on SICI vs SAI and LAI indicates that they are mediated by GABAergic circuits? Could it be opposite effect on the same circuit?

To improve the clarity of the text, we have now removed this sentence from the main text.

Lines 404-405 - "MEP amplitude only increase a few tens of milliseconds.." Other studies have found different results with MEP amplitude increase occurring earlier (e.g. PMID: 16932969, PMID: 34234660, PMID: 9749597). While different experimental design and analyzes may account for some of the differences, these results should be mentioned.

Fixed.

Lines 462-462 - It is unclear what type of future work the authors are suggesting & Lines 468-469 - What are the "advanced neuromodulation techniques"?

We thank the reviewer for bringing up these points of clarity. We have modified the text and removed these phrases, with the text now reading, "By designing adequately controlled experimental paradigms, it is possible to measure excitability changes in specific cortical pathways associated with changes in M1 and its inputs at different spatio-temporal scales. This information combined with population activity measured with neuroimaging has advanced our understanding of the cortical neural processes during motor control thanks to the complementarity of these different 'windows' into the brain. Future research combining TMS with invasive recordings will be critical for understanding how cellular, synaptic and neural population changes contribute to MEP amplitude changes in the context of movement."

Line 541 - Previous studies have shown steeper input-output relationship in Parkinson's disease (PD) at rest but less steep during muscle activation (PMID: 8164834).

We thank the reviewer for their suggestion to add this important study to our manuscript. The text now includes this article along with a description that "voluntary contraction of the muscle targeted with TMS significantly reduces the input-output relationship in PD..."

Lines 542-555 - Multiple studies have reported the abnormalities described by the authors. While it is not practical to cite all the studies, I believe that the first study that showed the abnormality should be referred to. For example, in PD decreased SICI was first described by Ridding et al (PMID: 7847860), increase in SICI with deep brain stimulation was first described by Cunic et al (PMID: 12058096) and increased SICF was first described by Ni et al. (PMID: 23576626).

We agree with the reviewer that the first studies should receive recognition. The important articles suggested by the reviewer have now been added to the manuscript.

Line 545 - A large study has shown that SICI is abnormal in the less affected hemisphere in PD (Ammann et al, PMID: 33141146).

We thank the reviewer for bringing this article to our attention, the text now reads "While these changes are thought to be most prominent in the hemisphere contralateral to the most affected body side (Spagnolo et al., 2013; (Kojovic et al., 2012), recent work with a large sample size of patients has also demonstrated that SICI is abnormal in the less affected hemisphere (Ammann et al., 2022)."

Line 653 - "reduce individual responsiveness to TMS" TMS responses when delivered at some EEG phases are increased rather than reduced (e.g. PMID: 31631058)

We thank the reviewer for pointing out the oversimplification in our text. We have modified the sentence as "...whether particular oscillations of synchronized brain activity can shape the output response of M1, which can be used to develop more precise stimulation protocols that are tailored to the individual's ongoing brain state (Hannah et al., 2016; Zrenner et al., 2020)."

Line 673 - "how activity spreads from M1 to other areas of the cortex". One of the advantages of

TMS-EEG compared to MEP measurement is that responses to TMS from cortical areas other than the M1 can be measured. This was not mentioned.

We thank the reviewer for bringing this to our attention. We have now added how “cortical areas other than M1 respond to TMS” as another advantage of TMS-EEG.

Line 701 - "diverse" should probably be "different"

Fixed

Line 707-708. The authors used "motoneuron" in the figure but "motor neuron" in the figure legend. Both can be used but the authors should use one term consistently.

Fixed

Fig 3A. The figure depicts LICI inhibiting SICI. This should be discussed and appropriate references cited.

We thank the reviewer for their important suggestion. In the caption, we now discuss a bit about the interactions of LICI and SICI and how this can be tested. The text now reads, “Moreover, the use of triple-pulse TMS protocols can be used to see how different neural populations interact with one another (e.g. LICI vs SICI). Using this strategy, LICI has been shown to reduce the effect of SICI (Sanger et al., 2001), a phenomenon that has been suggested to occur by presynaptic GABA_B receptor (LICI)-mediated inhibition (Sanger et al., 2001; McDonnell et al., 2006; Muller-Dalhaus et al., 2008).”

Fig 4. I find this figure confusing. What do the different colors for the pyramidal tract neurons represent? In (A) and (C), only one of the PTNs has connections to the interneurons but in (B) all 4 of them have connections - why? How the interneurons led to inhibition of all 4 PTNs in (A) and only one PTN in (B) are not clear. I do not see "different current directions" being depicted in (C).

We apologize for the confusion and lack of consistency surrounding Figure 4 and have now adjusted the image to display more consistency across each figure element. We hope that with the new design it is clear that in (A), there is a global inhibition that occurs across all PTNs when compared to (B), in which a specific PTN is inhibited. For part (C), we have now made two different scenarios (depicted by different colors), in which a particular network of neurons would be targeted with different current directions.

Dear Dr Spampinato,

Re: JP-TR-2023-281885XR1 "Motor potentials evoked by transcranial magnetic stimulation: interpreting a simple measure of a complex system" by Danny Adrian Spampinato, Jaime Ibanez Pereda, Lorenzo Rocchi, and John C Rothwell

Thank you for submitting your manuscript to The Journal of Physiology. It has been assessed by a Reviewing Editor and by 2 expert referees and we are pleased to tell you that it is acceptable for publication following satisfactory revision.

ABSTRACT FIGURES: Authors may use The Journal's premium BioRender account to create/redraw their Abstract Figures (and any other suitable schematic figure). Information on how to access this account is here: <https://physoc.onlinelibrary.wiley.com/journal/14697793/biorender-access>.

REVISION CHECKLIST:

We look forward to receiving your revised submission.

Yours sincerely,

Professor Laura Bennet
Senior Editor
The Journal of Physiology
<https://jp.msubmit.net>
<http://jp.physoc.org>
The Physiological Society
Hodgkin Huxley House
30 Farringdon Lane
London, EC1R 3AW
UK
<http://www.physoc.org>
<http://journals.physoc.org>

EDITOR COMMENTS

Reviewing Editor:

Both referees are in agreement that the manuscript has been improved markedly. A number of issues have however been highlighted which, if addressed, are likely to enhance the potential impact of the paper. These are detailed in the referees' comments. It would be particularly valuable if additional figures could be included - to illustrate key features of the phenomena under consideration.

REFeree COMMENTS

Referee #5:

The manuscript has improved significantly. Is still have a few comments:

- I recommended to not only list the used acronyms but also reduce their number throughout the text for readability, at least those that are note used frequently.
- After reading the text again carefully, I actually missed a bit the perspective of neuromodelling, which might deserve a few paragraphs. Capaday has just recently designed a rather beautiful simulation model that can generate MEPs including the entire chain from the cortex to the muscle, and there is more literature that models MEPs with and without variability. I found literature on size and mechanisms of variability. Although there are many modeling studies, some combined with measurements, none of them are discussed here although that branch seems very fruitful.
- Such modeling work also appears to shade more light on the cortical versus spinal variability question.
- Could the authors compare the variability of TEPs and MEPs quantitatively? Right now, it sounds like TEPs would be less variable. However, why does everyone then average dozens to hundreds of trials to measure TEPs?
- Also the authors did maybe not fully follow the question about the ISIs used in TEP measurement: For MEPs, most researchers use many seconds as large correlation was found for ISIs even exceeding 10 seconds (by the way, also not well covered in the text, which it should as such correlation effects are part of the variability). That finding is however ignored entirely in TEP measurements, which are recorded with ISIs lower than 2 seconds. I am not even aware of any study on the matter. I would expect this text to enlighten me. After all, it wants to be the go to paper on the topic.
- There is a lengthy discussion on the number of samples to be averaged to deal with the skewed distribution. How bad is the skewed distribution? Which shape does the distribution have? There is also notably more known already. Depending on how bad the skewness is, it might also need to be discussed if the average is a good way to describe the MEP size. Concentrating on the averages of a highly skewed distributions practically resulted in a worldwide financial crisis some 10 years ago. While economists were ridiculed by other disciplines in the aftermath for their previously poor statistical understanding, statisticians see the same problem in medicine, for example here. Instead of a discussion of how often to measure for an average, I would expect a discussion of the size of the variability, its distribution, in turn the skewness (quantitatively), and then on a more fundamental level if the average is good here or if the median might be better, for example. That might have more impact than many other aspects. After all, the MEP size is an important metric for detecting effects. If the current way of quantifying is not well defined, totally good studies become widely underpowered and results rather variable with the risk of statistical sampling artifacts. If you identify such a problem of measurement, it could solve some part of the high variability in MEP-dependent procedures and also reduce the reproducibility issue in our field so that we are left with physiological variability and do not add additional statistical sampling issues to it.
- Are there any histograms of MEPs that would really show the variability, maybe in combination with the traces on top of each other? It feels a bit like the elephant in the room: The text is dedicated to MEP variability, but not a single picture shows

it.

Referee #6:

The authors mostly addressed the specific comments raised. I have one minor comment.

The legend for Fig 4C stated that "When TMS is given with different current directions, changes in MEP amplitudes may result in less effective depolarization of intracortical neurons due to network properties". However, the figure shows that AP and PA currents recruit different sets of interneurons and activate different PTNs, but does not show one direction is more or less "effective" than the other one.

END OF COMMENTS

2nd Confidential Review

04-Jan-2023

The manuscript has improved significantly. It still has a few comments:

- I recommended to not only list the used acronyms but also reduce their number throughout the text for readability, at least those that are not used frequently.
- After reading the text again carefully, I actually missed a bit the perspective of neuromodelling, which might deserve a few paragraphs. Capaday has just recently designed a rather beautiful simulation model that can generate MEPs including the entire chain from the cortex to the muscle, and there is more literature that models MEPs with and without variability. I found literature on size and mechanisms of variability. Although there are many modeling studies, some combined with measurements, none of them are discussed here although that branch seems very fruitful.
- Such modeling work also appears to shed more light on the cortical versus spinal variability question.
- Could the authors compare the variability of TEPs and MEPs quantitatively? Right now, it sounds like TEPs would be less variable. However, why does everyone then average dozens to hundreds of trials to measure TEPs?
- Also the authors did maybe not fully follow the question about the ISIs used in TEP measurement: For MEPs, most researchers use many seconds as large correlation was found for ISIs even exceeding 10 seconds (by the way, also not well covered in the text, which it should as such correlation effects are part of the variability). That finding is however ignored entirely in TEP measurements, which are recorded with ISIs lower than 2 seconds. I am not even aware of any study on the matter. I would expect this text to enlighten me. After all, it wants to be the go to paper on the topic.
- There is a lengthy discussion on the number of samples to be averaged to deal with the skewed distribution. How bad is the skewed distribution? Which shape does the distribution have? There is also notably more known already. Depending on how bad the skewness is, it might also need to be discussed if the average is a good way to describe the MEP size. Concentrating on the averages of a highly skewed distributions practically resulted in a worldwide financial crisis some 10 years ago. While economists were ridiculed by other disciplines in the aftermath for their previously poor statistical understanding, statisticians see the same problem in medicine, for example here. Instead of a discussion of how often to measure for an average, I would expect a discussion of the size of the variability, its distribution, in turn the skewness (quantitatively), and then on a more fundamental level if the average is good here or if the median might be better, for example. That might have more impact than many other aspects. After all, the MEP size is an important metric for detecting effects. If the current way of quantifying is not well defined, totally good studies become widely underpowered and results rather variable with the risk of statistical sampling artifacts. If you identify such a problem of measurement, it could solve some part of the high variability in MEP-dependent procedures and also reduce the reproducibility issue in our field so that we are left with physiological variability and do not add additional statistical sampling issues to it.
- Are there any histograms of MEPs that would really show the variability, maybe in combination with the traces on top of each other? It feels a bit like the elephant in the room: The text is dedicated to MEP variability, but not a single picture shows it.

Dear Reviewers,

Thank you for the positive reviews and insightful feedback on our review article. By addressing the concerns step-by-step brought up by each reviewer, we believe the quality of our manuscript has noticeably improved. Below we provide responses to the concerns raised by the reviewers:

Referee #5:

The manuscript has improved significantly. It still has a few comments:

-1. I recommended to not only list the used acronyms but also reduce their number throughout the text for readability, at least those that are not used frequently.

We apologize for not removing acronyms that were seldom used. We have now reduced the number of acronyms to improve the readability of the articles.

-2.1. After reading the text again carefully, I actually missed a bit the perspective of neuromodelling, which might deserve a few paragraphs. Capaday has just recently designed a rather beautiful simulation model that can generate MEPs including the entire chain from the cortex to the muscle, and there is more literature that models MEPs with and without variability. I found literature on size and mechanisms of variability. Although there are many modeling studies, some combined with measurements, none of them are discussed here although that branch seems very fruitful. 2.2 Such modeling work also appears to shed more light on the cortical versus spinal variability question.

We thank the reviewer for their important suggestion in adding recent work that has begun to model MEP variability. As the reviewer suggested, we agree that adding some of the results and explanation of Capaday's model would be essential to inform future readers of this review article to gain some appreciation of cortical versus spinal influences on variability. Therefore, in the section "Measuring MEP: Variability," we have now added an entire paragraph that introduces and discusses the model by Capaday.

- 3. Could the authors compare the variability of TEPs and MEPs quantitatively? Right now, it sounds like TEPs would be less variable. However, why does everyone then average dozens to hundreds of trials to measure TEPs?

We thank the reviewer for bringing up this point of clarity. The reason tens/hundreds of responses need to be averaged to measure TEPs is that the signal-to-noise ratio of TEPs is quite low, limiting the reliability of single-trial responses. Collecting several responses is the standard approach for recording sensory-evoked potentials (e.g., visual, somatosensory, brainstem) (see <http://eknygos.lsmuni.lt/springer/586/485-497.pdf>). Moreover, we are unaware of any study that has attempted to compare the variability of TEPs and MEPs, but this comparison might not

make much sense since the reason for collecting more samples is quite different between TEPs and MEPs.

- 4. Also the authors did maybe not fully follow the question about the ISIs used in TEP measurement: For MEPs, most researchers use many seconds as large correlation was found for ISIs even exceeding 10 seconds (by the way, also not well covered in the text, which it should as such correlation effects are part of the variability). That finding is however ignored entirely in TEP measurements, which are recorded with ISIs lower than 2 seconds. I am not even aware of any study on the matter. I would expect this text to enlighten me. After all, it wants to be the go to paper on the topic.

We apologize for not precisely following the reviewer's initial concern. In the main text, we have added that it is essential to consider that the inter-pulse interval between pulses is an important experimental factor that influences MEP variability (under the section "Measuring MEP: variability").

Regarding inter-pulse intervals and TEPs, studies typically use ~2 seconds between pulses, mainly for practical reasons. As our previous response (comment #3) mentioned, several pulses are needed to achieve a reliable TEP signal. If one were to use a long interval between pulses (something like 10 seconds), the experimental protocol could take a very long time to perform, which could introduce various problems for extracting a reliable signal (like subject fatigue over the course of the experiment). No direct study has compared TEP responses that were done in short intervals (~2 seconds) versus long intervals (~10 seconds) as has been done for MEPs; however, a recent report did find that short ISIs do not cause problems with TEP signals, at least within the same block. In this study, the authors measured the TEP amplitude trend during a block of 100 trials delivered with IPIs jittering between 1.1 and 1.4 s randomly, thus leading the authors to argue that short IPIs do not affect TEP size and do not lead to any short-term plasticity.

On the other hand, Casarotto et al., 2022, recently developed a method that can do real-time monitoring of the data quality of TEPs. This would allow the experimenter to modify stimulation parameters based on a direct functional readout from the stimulated brain area. One application of this could be an experiment that tests whether TEP responses are sensitive to different ISIs.

-5. There is a lengthy discussion on the number of samples to be averaged to deal with the skewed distribution. How bad is the skewed distribution? Which shape does the distribution have? There is also notably more known already. Depending on how bad the skewness is, it might also need to be discussed if the average is a good way to describe the MEP size. Concentrating on the averages of a highly skewed distributions practically resulted in a worldwide financial crisis some 10 years ago. While economists were ridiculed by other disciplines in the aftermath for their previously poor statistical understanding, statisticians see

the same problem in medicine, for example here. Instead of a discussion of how often to measure for an average, I would expect a discussion of the size of the variability, its distribution, in turn the skewness (quantitatively), and then on a more fundamental level if the average is good here or if the median might be better, for example. That might have more impact than many other aspects. After all, the MEP size is an important metric for detecting effects. If the current way of quantifying is not well defined, totally good studies become widely underpowered and results rather variable with the risk of statistical sampling artifacts. If you identify such a problem of measurement, it could solve some part of the high variability in MEP-dependent procedures and also reduce the reproducibility issue in our field so that we are left with physiological variability and do not add additional statistical sampling issues to it.

We thank the reviewer for this vital comment to improve our manuscript. We have added a new paragraph in the “Measuring MEP: variability” section to address this concern. This paragraph includes potential issues and considerations when using the average to describe the MEP. It discusses crucial factors that can influence the MEP variability. It provides an example of how MEP distribution spread and skewness can change depending on where stimulation is given along the input-output curve.

-6. Are there any histograms of MEPs that would really show the variability, maybe in combination with the traces on top of each other? It feels a bit like the elephant in the room: The text is dedicated to MEP variability, but not a single picture shows it.

We thank the reviewer for this critical suggestion and agree that a figure depicting an example of variability would be helpful for readers of this review article. We have added a new figure to the manuscript (Figure 3) showing the MEP responses of three participants in different TMS state conditions, including their MEP traces and distribution of responses. This will clearly show the reader that various potential distributions exist across individuals and that changing the experimental condition may also influence the skewness of the distribution, thus demonstrating within- and across-subject variability.

Referee #6:

The authors mostly addressed the specific comments raised. I have one minor comment.

The legend for Fig 4C stated that "When TMS is given with different current directions, changes in MEP amplitudes may result in less effective depolarization of intracortical neurons due to network properties". However, the figure shows that AP and PA currents recruit different sets of interneurons and activate different PTNs, but does not show one direction is more or less "effective" than the other one.

We thank the reviewer for pointing out this discrepancy between the text and visualization of this figure (of note, now Figure 5). The image now displays that AP currents are less effective (i.e., by depicting fewer neurons stimulated) when compared to PA currents.

Dear Dr Spampinato,

Re: JP-TR-2023-281885XR2 "Motor potentials evoked by transcranial magnetic stimulation: interpreting a simple measure of a complex system" by Danny Adrian Spampinato, Jaime Ibanez Pereda, Lorenzo Rocchi, and John C Rothwell

We are pleased to tell you that your paper has been accepted for publication in The Journal of Physiology.

Authors should note that it is too late at this point to offer corrections prior to proofing. The accepted version will be published online, ahead of the copy edited and typeset version being made available. Major corrections at proof stage, such as changes to figures, will be referred to the Editors for approval before they can be incorporated. Only minor changes, such as to style and consistency, should be made at proof stage. Changes that need to be made after proof stage will usually require a formal correction notice.

All queries at proof stage should be sent to: TJP@wiley.com

Yours sincerely,

Professor Laura Bennet
Senior Editor
The Journal of Physiology
<https://jp.msubmit.net>
<http://jp.physoc.org>
The Physiological Society
Hodgkin Huxley House
30 Farringdon Lane
London, EC1R 3AW
UK
<http://www.physoc.org>
<http://journals.physoc.org>

P.S. - You can help your research get the attention it deserves! Check out Wiley's free Promotion Guide for best-practice recommendations for promoting your work at www.wileyauthors.com/eoo/guide. You can learn more about Wiley Editing Services which offers professional video, design, and writing services to create shareable video abstracts, infographics, conference posters, lay summaries, and research news stories for your research at www.wileyauthors.com/eoo/promotion.

IMPORTANT NOTICE ABOUT OPEN ACCESS: To assist authors whose funding agencies mandate public access to published research findings sooner than 12 months after publication The Journal of Physiology allows authors to pay an Open Access (OA) fee to have their papers made freely available immediately on publication.

You can check if your funder or institution has a Wiley Open Access Account here: <https://authorservices.wiley.com/author-resources/Journal-Authors/licensing-and-open-access/open-access/author-compliance-tool.html>

EDITOR COMMENTS

Reviewing Editor:

Thank you for addressing the comments provided in the most recent round of reviews.

REFeree COMMENTS

Referee #8:

The authors have addressed the minor issue raised in the previous review.

3rd Confidential Review

30-Mar-2023